

# Distributed summer air temperatures across mountain glaciers: climatic sensitivity and glacier size

Thomas E. Shaw[1], Wei Yang[2,3], Álvaro Ayala[4], Claudio Bravo[5], Chuanxi Zhao[2], Francesca Pellicciotti[6,7]

[1] Advanced Mining Technology Center, Universidad de Chile, Santiago, Chile
[2] Key Laboratory of Tibetan Environment Changes and Land Surface Processes, Institute of Tibetan Plateau Research, Chinese Academy of Sciences (CAS), Beijing, China
[3] CAS Center for Excellence in Tibetan Plateau Earth Sciences, Beijing 100101, China
[4] Centre for Advanced Studies in Arid Zones (CEAZA), La Serena, Chile
[5] School of Geography, University of Leeds, Leeds, UK
[6] Federal Institute for Forest, Snow and Landscape Research (WSL), Birmensdorf, Switzerland
[7] Department of Geography, Northumbria University, Newcastle, UK

*Corresponding author: Thomas E. Shaw (thomas.shaw@amtc.uchile.cl)*

Keywords: Air Temperature, Glaciers, Tibetan Plateau, Climatic Sensitivity

## Abstract

Near-surface air temperature ($T_a$) is highly important for modelling glacier ablation, though its spatio-temporal variability over melting glaciers still remains largely unknown. We present a new dataset of distributed $T_a$ for three glaciers of different size in the south-east Tibetan Plateau during two monsoon-dominated summer seasons. We compare on-glacier $T_a$ to ambient $T_a$ extrapolated from several, local off-glacier stations. We parameterise the along-flowline climatic sensitivity of $T_a$ on these glaciers to changes in off-glacier temperatures and present the results in the context of several available distributed on-glacier datasets around the world. Climatic sensitivity decreases rapidly up to 2000-3000 m along the down-glacier flowline distance. Beyond this distance, both the $T_a$ of the Tibetan glaciers and global glacier datasets show a slower decrease of climatic sensitivity. In general, observations on small glaciers (with < 1000 m flowline distance) are highly sensitive to temperature changes outside the glacier boundary layer. The climatology of a given region can influence the general magnitude of this climatic sensitivity, though no strong relationships are found between along-flowline climatic sensitivity and mean summer temperatures or precipitation. The terminus of some glaciers remain associated with other warm air processes that increase climatic sensitivity (such as divergent boundary layer flow, warm up-valley winds or debris heating effects) which are evident only beyond ~70% of the total glacier flowline distance. Our results therefore suggest a strong role of local effects in modulating climatic sensitivity close to the glacier terminus, although further work is still required to explain the variable presence of these effects for different glaciers.

## 1. Introduction

Near-surface air temperature ($T_a$) is one of the dominant controls on glacier energy and mass balance during the ablation season (Petersen et al., 2013; Gabbi et al., 2014; Sauter and Galos, 2016; Maurer et al., 2019; Wang et al., 2019), though modelling its spatio-temporal behaviour above melting ice surfaces remains a challenge. The absence of distributed information regarding $T_a$ has favoured the use of simple, space-time invariant relationships of $T_a$ with elevation, typically



that of the free-air environmental lapse rate (*ELR*). The physical processes of the free-air (that
which is independent of the surface boundary layer), however, are not appropriate to describe the
variability of $T_a$ for local glacier boundary layers (Figure 1a), especially when the above-ice
temperature gradient (within ~10 m of the ice surface) heightens under warm 'ambient' (off-
glacier) conditions (van den Broeke, 1997; Greuell and Böhm, 1998; Oerlemans, 2001;
Oerlemans and Grisogono, 2002; Ayala et al., 2015). As a result, any extrapolation of $T_a$
observations from an off-glacier location, particularly those at lower elevations, are likely to lead
to an overestimation of snow and ice ablation in melt simulations (Petersen and Pellicciotti, 2011;
Pellicciotti et al., 2014; Shaw et al., 2017). Whilst this problem has been long understood (Greuell
et al., 1997; Greuell and Böhm, 1998), only within the last decade have studies approached it in
more detail (Petersen et al., 2013; Ayala et al., 2015; Carturan et al., 2015; Shaw et al., 2017;
Bravo et al., 2019; Troxler et al., 2020). Until recently, modelling studies have relied upon simple
lapse rates (including the *ELR*) and/or single bias offset values to account for the 'cooling effect'
of the near-surface air on-glacier (Arnold et al., 2006; Nolin et al., 2010; Ragettli et al., 2016).
The variations of $T_a$ along the glacier flowline (defined following Shea and Moore (2010) as the
horizontal distance from an upslope summit or ridge), however, are much more complex (Ayala
et al., 2015; Shaw et al., 2017), though a lack of available data usually restricts one's ability to
appropriately model this variable. While models applying the degree day approach can make use
of off-glacier temperatures as forcing because they are heavily reliant on calibration, for
physically based models and models of intermediate complexity (Pellicciotti et al., 2005; Ragettli
et al., 2016) it is key to resolve the air temperature distribution over glaciers, especially for
turbulent flux calculations.
To date, two main, simplified model approaches have been developed and tested to represent air
temperature over glaciers (Figure 1a). The first is the statistical model by Shea and Moore (2010)
developed to reconstruct $T_a$ across glaciers of varying size in the Canadian Rockies from ambient
temperature records. This approach considered the ratio of observed on-glacier temperature and
estimated ambient temperature for the elevation of a given point (hereafter '$T_aAmb$') above and
below a critical threshold temperature for the onset of the glacier katabatic boundary layer (*KBL*).
The parameterisations that operate as a function of the along-flowline distance have since been
explored by Carturan et al. (2015) and Shaw et al. (2017) on smaller glaciers in different parts of
the Italian Alps. Carturan et al. (2015) found that the original published parameterisations were
sufficient to explain $T_a$ on small, fragmenting glaciers up to distances of ~2000m. However,
investigation by Shaw et al. (2017) on a small alpine glacier found a pattern of along-flowline $T_a$
that was better described by an alternative, thermodynamic model approach. This second,
physically-oriented approach was developed by Ayala et al. (2015) based upon modifications of
the original model by Greuell and Böhm (1998) to account for a relative 'warming effect' evident
on the termini of some mountain glaciers compared to upper elevations that were fully dominated
by katabatic winds. The modified model (termed 'ModGB' in the literature) accounts for the
down-glacier cooling of $T_a$ at increasing flowline distances due to sensible heat exchange and
adiabatic heating (Greuell and Böhm, 1998). It adds, however, an additional warming factor based
upon on-glacier observations in the lower sections of the glacier (e.g. at the greatest flowline
distances) to account for additional processes of adiabatic warming (Ayala et al., 2015) (Figure
1a). The ModGB approach has been successively applied at other glacier sites around the world
(Shaw et al., 2017; Troxler et al., 2020), though the question of its transferability remains open
(Troxler et al., 2020).
Thus the ModGB method operates on the physical principles of the glacier boundary layer
(Greuell and Böhm, 1998) though it corrects for relative warming on the lower portion of glacier
(Ayala et al., 2015). To establish the magnitude of this warming, however, along-flowline data in
the lower portion of the glacier are essential. Because the available distribution of on-glacier
observations is often limited and rarely extends for the entire length of the glacier boundary layer,
this additional correction for warming and the number of physical unknowns of ModGB can lead
to high variability in $T_a$ estimates on the glacier terminus (Troxler et al., 2020) (Figure 1a). In
contrast to this, the statistical method of Shea and Moore (2010) provides a more simplified



estimation that has fewer assumptions and parameters, though it does not explicitly account for
the physical processes on the glacier, especially those that are thought to be the cause of relative
warming for the glacier terminus. It also provides a parameter that more specifically represents
the glacier 'climatic sensitivity' of the on-glacier $T_a$ (defined here as the ratio of changes in
observed $T_a$ on-glacier to changes in $T_aAmb$). Despite its more conceptual nature, because of its
greater generalisability typical of a more simplistic statistical approach, we adopt the Shea and
Moore (2010) method to further investigate along-flowline $T_a$ in this study.
To the author's knowledge, no study has investigated the variability of on-glacier $T_a$ at different
sites around the world (with the exception of three glaciers considered by Ayala et al., (2015)).
As such the transferability or generalisability of models and/or model parameters remain mostly
unknown, and analysis of individual glacier sites, while beneficial to process understanding, may
not advance the science on how to treat the on-glacier $T_a$ in models.  In this study, we make a step
toward this by utilising new datasets of on-glacier temperature observations on three glaciers of
varying size in the south-east Tibetan Plateau. We analyse the main controls on along-flowline $T_a$
and its climatic sensitivity and present these new findings in the context of 11 other distributed
on-glacier observations around the world made to date.
Specifically we aim to i) understand the variability of $T_a$ with the along-flowline distances at three
glaciers in the south-east Tibetan Plateau, ii) identify and quantify the climatic sensitivity of on-
glacier $T_a$ for different meteorological conditions and glacier sizes and iii) parameterise the along-
flowline $T_a$ using the Shea and Moore (2010) method for the Tibetan glaciers and discuss it in the
context of globally-derived, published datasets of on-glacier air temperatures.

**2.  Study Site**
The study glaciers are located in the upper Parlung-Zangbo River catchment in the southeast Tibet
Plateau (29.24°N, 96.93°E - Figure 2), a region characterised by a summer monsoon climate that
typically intrudes via the Brahmaputra Valley (Yang et al., 2011). We present data for three
maritime-type valley glaciers in the wider Parlung catchment: Parlung Glacier Number 4
(hereafter 'Parlung4'), Parlung Glacier Number 94 ('Parlung94') and Parlung Glacier Number
390 ('Parlung390'). Parlung4 (Figure 2d) is ~10.8 km², north-northeast facing and has an
elevation range of 4659-5939 m a.s.l. (Ding et al., 2017). Glaciers Parlung94 (Figure 2c) and
Parlung390 (Figure 2e) are smaller valley glaciers (2.51 and 0.37 km², respectively) that have
termini at higher elevations (elevation ranges of 5000-5635 and 5195-5469 m a.s.l.,
respectively).  The glaciers of the catchment were classified by Yang et al. (2013) as having a
spring-accumulation regime and the largest annual rain season of the entire Tibetan Plateau. The
upper Parlung River catchment has a mean summer (1979-2019) annual air temperature of ~2°C
(at 4600 m a.s.l.), and temperatures in the wider region have been shown to be increasing since
the mid 1990's (Yang et al., 2013). The glaciers of this region have been shown to be very
sensitive to temperature changes, though with a more elevation-independent mass balance
sensitivity compared to other, continental glaciers of the Tibetan Plateau (Wang et al., 2019). The
accurate estimation of on-glacier temperatures as Tibetan glaciers shrink and fragment (Carturan
et al., 2015) is thus of significant importance for continued modelling efforts. However, to date,
no such studies regarding on-glacier temperature distribution have performed within the Tibetan
Plateau.
**3. Data**
149        *3.1.    Air temperature observations*
We present the observations of $T_a$ from a total of 20 air temperature logger locations (Table 1),
13 of which are situated on-glacier (4680 - 5369 m a.s.l.) and seven off-glacier (4648 - 5168 m
a.s.l.). These stations (hereafter referred to as 'T-loggers') observed $T_a$ at a 2 m height using
HOBO U23-001 temperature-relative humidity sensors (accuracy +0.21°C) within double-
louvered, naturally-ventilated radiation shields mounted on free-standing tripods. The T-loggers
recorded data in 10 minute intervals that are averaged to hourly data for analysis. We identify a
common observation period over the summers of 2018 and 2019 that range from 12th July – 18th





September. For these date ranges, we observe only small data gaps for some T-loggers (Table 1).
We apply the nomenclature of T$X_G$, whereby $X$ refers to the T-logger number on each glacier and
$G$ refers to the glacier number.
We additionally present $T_a$ observations at two automatic weather stations (AWS) at elevations
~4600 m a.s.l. (off-glacier, henceforth 'AWS_Off') and ~4650 m a.s.l. (on Parlung4, henceforth
'AWS_On') for the same time period (Figure 2). For distributing off-glacier air temperature, we
consider AWS_Off as our reference station. The AWS $T_a$ observations are provided by Vaisala
HMP60 temperature-relative humidity sensors (accuracy +0.5°C) also housed in naturally-
ventilated radiation shields.

### 3.2.   Uncertainty of air temperature observations

To provide an estimate of observation uncertainty, we compared the hourly divergence of two
naturally-ventilated $T_a$ observations for the whole period between T4$_4$ and AWS_On (Figure 2d),
that are co-located within a few metres of horizontal distance on Parlung4 Glacier. A test of
absolute differences between the two stations resulted in a mean of < 0.4°C for all hours (n =
3312) and ~0.5°C for the warmest 10% of the hours of ambient temperature at AWS_Off
(hereafter referred to as 'P90' - (Ayala et al., 2015; Shaw et al., 2017; Troxler et al., 2020)).  We
find that for these hours (when the *KBL* development is theoretically at its strongest (e.g. van den
Broeke, 1997; Oerlemans and Grisogono, 2002)), that 95% of hourly differences were < 1°C
(Figure S1). For on-glacier stations at large flowline distances (Figure 2), these large uncertainties
are considered less likely given the good ventilation provided to the sensors within the
*KBL*. While observations at short flowline distances with calm conditions and high incoming
radiation may result in maximum uncertainties up to ~1°C (Troxler et al., 2020), we apply a
±0.5°C uncertainty for analysis of distributed $T_a$. For the instantaneous differences > 1°C, wind
speeds at AWS_On were <2 m s$^{-1}$. Wind speeds for P90 conditions were otherwise in excess of
3-4 m s$^{-1}$, though no other observations of on-glacier wind speed are available at higher elevations.

### 3.3.   Meteorological information

We obtained information regarding $T_a$, incoming shortwave radiation and relative humidity
(AWS_Off), on-glacier wind speed (AWS_On) and 'free-air' wind speed and direction (ERA5 -
C3S, 2017). We used these data to explore the relationships of hourly on- and off-glacier
temperatures (section 4.2) for different prevailing conditions.

### 3.4.   Elevation information

We used the 12.5 m Alos Palsar (ASF DAAC, 2020) digital elevation model (DEM) to provide
elevation information for the catchment (Figure 2b). We utilised this DEM in order to calculate
flowline distances (m) for each glacier from the TopoToolbox functions in Matlab (Schwanghart
and Kuhn, 2010), following Troxler et al, (2020). We note that the methodology for flowline
generation is not currently uniform among all studies of this type (Shea and Moore, 2010; Ayala
et al., 2015; Carturan et al., 2015; Shaw et al., 2017; Bravo et al., 2019; Troxler et al., 2020) and
may produce some differences in the calculated distances close to the lateral borders of the
glaciers. In addition, the generated flowlines may also be dependent upon the quality and
resolution of the DEM available between the aforementioned studies. However, we do not
analyse lateral $T_a$ variations in this study and consider that the impact of varying methods for
flowline generation to be negligible when assessing observations at a few select points on the
glacier.

## 4. Methods

For this study we use local, off-glacier $T_a$ data from AWS_Off for aggregation of on-glacier sub-
groups or for distribution of $T_a$ in space. Sub-grouping allows one to interpret general causal





factors that dictate on-glacier behaviour, whereas the distribution in space allows a direct
comparison of on- and off-glacier temperatures and the effect of the glacier boundary layer. The
following subsections outline the sub-grouping (4.1) and distribution (4.2) methodologies. The
model parameterisations of Shea and Moore (2010) and application to Tibetan and global datasets
are considered in sections 4.3 and 4.4, respectively.
*4.1.    Sub-grouping on-glacier air temperature observations*
We sub-group our on-glacier observations by 10th and 90th percentiles (P10/P90) of off-glacier
$T_a$ at AWS_Off (Figure 2a) that have been shown to relate to the development of the glacier
boundary layer (Ayala et al., 2015). Following the methodology of previous studies (Ayala et al.,
2015; Shaw et al., 2017; Troxler et al., 2020), we consider all contemporaneous observations of
on-glacier $T_a$ at each T-logger that relate to the same hours as the P10/P90 classification at
AWS_Off. We consider the deviation from a linear relationship of $T_a$ with elevation and flowline
distance for these subgroups, assessing this 'linearity' by use of the coefficient of determination
($R^2$). For a comparison to previous studies (Petersen and Pellicciotti, 2011; Shaw et al., 2017), we
also report the equivalent on-glacier lapse rate that would be calculated for the above conditions.
*4.2.    Comparison of on- and off-glacier air temperature*
We extrapolate AWS_Off $T_a$ records to the elevation of each on-glacier T-logger (Table 1) to
quantify the $T_a$ differences within the glacier boundary layer (Figure 1a). We derive an hourly
variable lapse rate between AWS_Off and off-glacier T-loggers deemed to be independent of the
glacier wind layer, thus excluding those T-loggers in the immediate pro-glacial zones.
Specifically, we use AWS_Off and T-loggers $T1_{94}$, $T2_{94}$ and $T1_{390}$ to construct a 'catchment lapse
rate' where the origin of the calculated regression must pass through the elevation of AWS_Off
that acts as the forcing station in this study (see supplementary information, Figure S2). We
consider this as the best available approach to estimate the ambient lapse rate for the catchment.
We compare the hourly estimates of extrapolated off-glacier $T_a$ ($T_aAmb$) with the observations at
each on-glacier T-logger in order to i) understand how large the $T_a$ offset (bias) is at each site and
how it relates to meteorological conditions and glacier flowline distance; and ii) parameterise the
along flowline climatic sensitivity to $T_aAmb$ following Shea and Moore (2010) (section 4.3).
*4.3.    Estimation of on-glacier climatic sensitivity*
The Shea and Moore (2010) approach (hereafter 'SM10') estimates on-glacier $T_a$ using $T_aAmb$ at
a given elevation by:

$$Ta = \begin{cases} T1 + k2(T_aAmb - T*), & T_aAmb \geq T* \\ T1 - k1(T* - T_aAmb), & T_aAmb < T* \end{cases}$$

*(1)*

where $T*$ (°C) represents the threshold ambient temperature for the onset of katabatic flow and
$T1$ is the corresponding threshold $T_a$ on the glacier. Parameters $k1$ and $k2$ are the climatic
sensitivities of on-glacier $T_a$ to $T_aAmb$ below and above the threshold $T*$ (Figure 1b and c). $k1$
and $k2$ were parameterised in the original study using exponential functions of the along flowline
distance (*DF*):

$$k1 = \beta1 \exp(\beta2\, DF)$$
*(2)*

$$k2 = \beta3 + \beta4 \exp(\beta5\, DF)$$

*(3)*

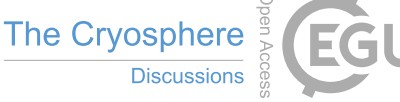

where $\beta i$ are the fitted coefficients. Following the suggestion of Carturan et al. (2015), we
implement a relation against the flowline that estimates the threshold temperature for onset of
katabatic effects ($T^*$) at a given distance as:

$$T* = \frac{C1 DF}{C2 + DF}$$

263                                                                                              *(4)*


where $C1$ (6.61) and $C2$ (436.04) are the fitted coefficients of Carturan et al. (2015). We calculate
$k1$ and $k2$ at each T-logger station using the linear regression of observed $T_a$ and $T_aAmb$ above
and below $T^*$ (Figure 1) as derived from equation 4. We note that the parameter $k2$ holds a greater
significance for modelling $T_a$ (Figure 1a), as this more closely represents the climatic sensitivity
reported by previous works (Greuell et al., 1997; Greuell and Böhm, 1998; Oerlemans, 2001;
2010), whereas $k1$ represents the ratio of above-glacier and free-air temperatures without a
katabatic effect that have been shown to relate more closely to $T_aAmb$ (Shea and Moore, 2010;
Shaw et al., 2017). For this study, we therefore pay particular attention to the $k2$ sensitivities on
the Parlung glaciers and assess their relationship to along-flowline distance.

*4.3.    Global datasets of on-glacier temperatures*
To explore the generalisability of the SM10 approach and provide context to the findings of the
Parlung catchment, we explore the calculated $k1$ and $k2$ parameters for several of the available
distributed on-glacier datasets published to date (Figure S3, Table 2). We subset summer periods
to when all available on-glacier observations are available at a given site. For sites of the Coastal
Mountains of British Columbia ('CMBC' - Shea and Moore, 2010) and Alta Val de La Mare
('AVDM' - Carturan et al., 2015), we apply the published parameter sets derived from those
authors. For all other sites, we derive $T_aAmb$ from the most locally available off-glacier AWS and
the published lapse rate from the relevant studies (Table 2). In the absence of lapse rate
information, we apply the *ELR* (-6.5°C km⁻¹) to extrapolate $T_a$ to the elevation of the on-glacier
observations.
For each glacier site, data are limited to those hours when all stations for that glacier are available
and the $k1$ and $k2$ parameters (equation 1) are only calculated when; i), >10% of the total hourly
data at a given station is above or below $T^*$ (to have enough data to calculate $k2$ and $k1$,
respectively) and, ii) the linear regression to derive each parameter is significant to the 0.95 level.
For those on-glacier stations that do not satisfy the above requirements, we do not calculate the
$k1$ and $k2$ parameters.

Finally, we group the derived $k2$ sensitivities of the SM10 approach against the climatology that
describes the given glacier(s) location. For this, we consider the mean summer (JJAS or DJFM in
the southern hemisphere) air temperature (MSAT) and the total annual precipitation for the year(s)
of study at each location (Table 2). MSAT is derived from the ERA5 product for the glacier
centroid location and corrected to the mean glacier elevation by the *ELR*. However, total
precipitation from ERA5 has been shown to have considerable bias when tested against in-situ
observations (e.g. Betts et al., 2019), and so we provide the best available value from the relevant
literature (Table 2). We note that a full analysis of the local climate is beyond the scope of this
work, though we attempted a generalised analysis in order to link any clear differences in the
global datasets to climatological influences.

**5. Results**
*5.1.    Variability of on-glacier air temperatures*
Figure 3 shows the mean $T_a$ as a function of elevation and flowline distance for the Parlung
glaciers for all conditions and for the warmest 10% of AWS_Off observations (P90). The average



of all hours reveals a generally linear relationship with the glacier elevation (Figure 3a) and
flowline (Figure 3b), resulting in mean on-glacier lapse rate (mean $R^2$ with elevation) equivalent
to -3.0°C km$^{-1}$ (0.92), -3.7°C km$^{-1}$ (0.71) and -4.5°C km$^{-1}$ (0.81) for Parlung4, Parlung94 and
Parlung390, respectively. For P90 hours (n = 312), mean $T_a$ demonstrates a poorer fit to elevation
and flowline for Parlung4 (mean $R^2$ with elevation = 0.12, and flowline = 0.20) and Parlung 94
(mean $R^2$ with elevation = 0.13 and flowline = 0.09). For the small Parlung390 Glacier, $T_a$ remains
strongly related to elevation ($R^2$ = 0.84) and flowline ($R^2$ = 0.82) under P90 conditions. The
equivalent mean on-glacier 'lapse rates' for P90 hours are -2.1°C km$^{-1}$, -1.4°C km$^{-1}$ and -4.1°C
km$^{-1}$. Nevertheless, assuming a calculated 0.5°C uncertainty of the observations for P90
conditions (Figure 3c and d), the mean of observations still lies along a linear fit line. However,
for given hours, the deviation of observations from the linear fit line exceeds 3°C at large flowline
distances (> 7000 m) on Parlung4. In general, 2018 experienced cooler average temperatures at
higher elevations, but in general, there are no marked differences between the two years of
observation when comparing to glacier elevation or flowline (not shown).
*5.2.    Differences in on- and off-glacier air temperatures*
Comparing mean on- and off-glacier $T_a$ reveals the expected behaviour associated with the glacier
'cooling effect' (Carturan et al., 2015) and a greater deviation from the calculated catchment lapse
rate for the warmest conditions (P90, Figure 4), indicating a reduced climatic sensitivity. The
mean $T_a$ observed at off-glacier T-Loggers supports the selection of those stations used for
catchment lapse rate calculation (green dots in Figure 4) that are further from the potential effects
of the glacier boundary layer (red markers in Figure 4). Following Carturan et al. (2015), we
suggest a potential non-linear behaviour of lapse rates between AWS_Off and the top of the
flowline for Parlung390, though we lack the off-glacier observations above the flowline origin to
test this (Figure 4b). We therefore utilise a piecewise lapse rate at the point of the highest off-
glacier lapse rate station (T1$_{390}$ - red line in Figure 4) to account for the discrepancy between the
estimated and observed $T_a$ at T6$_{390,}$ which is assumed to be near to the flowline origin where
climatic sensitivity is theoretically equal to 1 (i.e. that on-glacier observations = $T_a amb$).
Figure 5 presents the hourly differences between $T_a Amb$ and observed $T_a$ at each site. The
deviation of estimated and observed $T_a$ theoretically begins at a critical temperature threshold, $T^*$
(Shea and Moore, 2010) and this effect can be observed at T-logger sites on Parlung94 and
Parlung4, particularly those at greater flowline distances. Coloured by the hourly wind speeds
recorded at AWS_On, the beginning of the temperature deviations ($T^*$) aligns well with the onset
of katabatic winds on Parlung4 (and only assumed for the other glaciers due to lack of on-glacier
wind observations – Figure 5). Despite being pro-glacial stations, T1$_4$ and T2$_4$ reveal a similar,
albeit weaker effect of the glacier boundary layer, possibly due to larger glacier flowline and
sustained effect of the katabatic wind into the pro-glacial area.
The mean bias offset of along-flowline $T_a$ using the catchment lapse rate is shown in Figure 6.
For the coolest 10% of hours at AWS_Off (P10), there is generally minimal offset between $T_a Amb$
and observed $T_a$ for the entire dataset. This clearly does not hold true for P90 conditions (Figure
6a), as already established (Figure 4), and offsets of $T_a$ ($T_a Amb$ – observed $T_a$) are up to 5.8°C at
flowline distances of > 7000 m on Parlung4. These effects appear to heighten beyond 2000 m
along the flowline (Parlung94), though slight offsets can be witnessed for all glaciers. This is
generally associated with drier conditions, and for hours of greater relative humidity (AWS_Off),
offsets are small (Figure 6b). Considering 'free-air' wind variability provided by ERA5
reanalysis, $T_a$ offsets are largest for the dominant south-westerly wind direction (85% of hours)
and when free-air wind speeds are smallest (Figure 6c and d). However, un-corrected, gridded
wind speeds do not appropriately represent the local 'free-air' boundary conditions and thus the
interaction of off-glacier wind speeds and the glacier boundary layer development remain unclear
for these glaciers. For all but the coolest ambient temperatures (Figure 6a), observations at the
greatest flowline distances deviate the most from the estimated values.

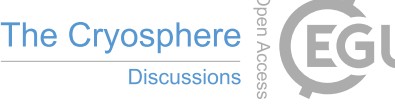

This offset is highly variable in time, however, and related to the prevailing conditions of a given
year (Figure 7). Considering the maximum daily $T_a$ offsets at the on-glacier T-Logger closest to
the terminus on each glacier (Table 1), we find that Parlung94 and Parlung4 T-loggers have
similar magnitudes of $T_a$ offsets during the mid-summer months, particularly for 2018 (Figure 7).
These maximum offsets are in clear relation to the incoming shortwave radiation record at
AWS_Off (correlations of 0.44, 0.60 and 0.80 for Parlung390, Parlung94 and Parlung4,
respectively), which are indicative of warmer ambient conditions (i.e. P90). For Parlung390 this
offset is much smaller, though varies considerably throughout the summer. For 2019, maximum
daily $T_a$ offsets on Parlung390 steadily increase during July and August then fall close to zero in
September. The bias offsets for Parlung4 and Parlung94, however, remain sizeable (Figure 7).
Because our study period focuses on the core monsoon period (Yang et al., 2011), we do not
observe the influence of monsoon arrival or cessation on the $T_a$ variability of the Parlung Glaciers.
*5.3.     Parameterisation of along-flowline air temperatures*
Figure 8 presents the SM10 parameterisations for the Parlung glaciers in comparison to those
derived for the available distributed $T_a$ datasets around the world (Table 2). Comparing the *k1* and
*k2* parameters from Tibet to the original parameters of Shea and Moore (2010), a similar
behaviour is observable for both sites up to ~2000-3000 m of flowline distance (red and blue
dashed lines), though there exists a larger variability in the calculated parameters at longer
flowline distances on Parlung4 (Figure 8). Accordingly, the exponential functions that are fitted
to the observations at Parlung glaciers and the original study are notably distinct (Figure 8, Table
3). This behaviour is further highlighted when observing other published or revised datasets for
the context of this work (Figure 8b). A 'global' parameterisation for all sites where down-glacier
decrease in climatic sensitivity is evident (black dashed lines in Figure 8) clearly misrepresents
many of the observations, particularly those at greater flowline distances, balancing the
behaviours reported for different sites.
Notably, observations at McCall Glacier, Alaska relate very well to ambient $T_a$ under cooler
conditions, with all *k1* values remaining > 0.9. Above the $T*$ threshold, however, the relationship
of observed and estimated $T_a$ results in increasing *k2* along the flowline, in contradiction to the
majority of the other datasets. Nevertheless, this data also confirms the increased climatic
sensitivity on the glacier terminus (Troxler et al., 2020) as evident with datasets for Tsanteleina
(Shaw et al., 2017), Arolla and Juncal Norte (Ayala et al., 2015). Observations at Parlung4 and
Universidad Glacier (Bravo et al., 2017) emphasise the strong decrease in climatic sensitivity at
large flowline distances (~10,000 m) previously only witnessed from one location on Bridge
Glacier, Canada (Shea and Moore, 2010).
Figure 9 shows the *k2* parameters plotted against flowline distance, coloured by rankings of
MSAT and precipitation totals (Table 2). The warmest of the investigation sites (during the
measurement years) appear to lie closer along the original SM10 parameterisation until ~4000 m,
whereas deviation of the *k2* parameters from this line appears larger for observations at relatively
cold sites (Greve, McCall and Arolla – Figure 9a). The main exception to this is for Juncal Norte
(Petersen and Pellicciotti, 2011; Ayala et al., 2015), which demonstrates a high and rapidly
increasing sensitivity of ambient $T_a$ at the greatest flowline distances.
No clear patterns are visible with relation to mean annual precipitation totals, although the
observations at Juncal Norte are noted as the driest of the study sites (Figure 9b).
A clear difference between observations of CMBC and Parlung at large flowline distances is the
distance from the glacier terminus, which suggests a possible difference in processes being
compared between sites. Accordingly, we plot the *k2* parameters as a function of the normalised
flowline, adjusted by the total length of glacier under the year(s) of observation (Table 2). A slight
trend toward lower *k2* values (lower sensitivity to ambient $T_a$) is observable for relatively warmer
regions, though no clear pattern emerges for MSAT (Figure 9c) or precipitation totals (Figure 9d).
The largest flowline distance observation of the entire dataset (Figure 9a) in fact extends only
~60% of the total glacier length (Bridge Glacier - CMBC), neither representing the smallest
climatic sensitivity (Figure 8b), nor the increasing climatic sensitivity witnessed at the terminus
of the glacier by other studies (Ayala et al., 2015). In fact, by subjectively grouping glacier sites




by the presence of the relative down-glacier cooling (decreasing sensitivity) and warming
(decreasing followed by increasing sensitivity) along the total glacier length, one can observe that
this effect is absent for the both smaller and larger glaciers (Figure 10a), albeit limited by lack of
observations across an entire glacier in most cases. For those glaciers where the pattern of down-
glacier sensitivity increase ('relative warming effect') is evident, it is found only beyond ~70%
of the total glacier flowline distance (Figure 10b – vertical dashed line). Up to this distance, no
increase in $k2$ sensitivity is seen from the data.

**6. Discussion**
426       *6.1.     Relevance of the findings from Parlung Glaciers*
The observations of along-flowline $T_a$ on the glaciers in the Parlung catchment add yet more
evidence of the spatial variability of the glacier cooling and dampening effect (Oerlemans, 2001;
Carturan et al., 2015; Shaw et al., 2017) and the need to appropriately estimate its behaviour for
use in glacier melt models (Petersen and Pellicciotti, 2011; Shaw et al., 2017; Bravo et al., 2019).
It has long since been observed that a static lapse rate is inappropriate for characterising the spatio-
temporal variability of $T_a$, both within the *KBL* (Greuell et al., 1997; Konya et al., 2007; Marshall
et al., 2007; Gardner et al., 2009; Petersen and Pellicciotti, 2011) and outside the glacier boundary
layer in adjacent valleys (Minder et al., 2010; Immerzeel et al., 2014; Gabbi et al., 2014; Heynen
et al., 2016; Jobst et al., 2016). Despite this, the lack of locally available observations often
requires modellers to force model simulations with the nearest off-glacier record of $T_a$ and
extrapolate it based upon the *ELR* value as a default. In the case of Tibetan glaciers, model studies
have often derived static lapse rates between on-and off-glacier stations (Huintjes et al., 2015) or
used static values to extrapolate or downscale $T_a$ with a correction based upon a single on-glacier
location (e.g. Caidong and Sorteberg, 2010; Yang et al., 2013; Zhao et al., 2014). To the author's
knowledge, this is the first time that such detailed information regarding spatio-temporal
variations in $T_a$ have been presented for a glacier of the Tibetan Plateau. Because glaciers of the
south-eastern Tibetan Plateau have been shown to be particularly susceptible to increases in $T_a$
(Wang et al., 2019), accurately parameterising $T_a$ along glaciers of differing size is highly relevant
for present and future melt modelling attempts. This is especially true where glaciers begin to
shrink or fragment (Munro and Marosz-Wantuch, 2009; Jiskoot and Mueller, 2012; Carturan et
al., 2015) and become more sensitive to ambient air temperatures due to a lack of katabatic
boundary layer development (Figures 6 and 7).

The summer monsoon exerts a strong control on the energy and mass balance of Tibetan glaciers
(Yang et al., 2011; Mölg et al., 2012; Zhu et al., 2015). Although our dataset spanned two
summers of only the core monsoon period for this region (Yang et al., 2011), we have shown that
the sensitivity of the glacier to external temperature changes (shown by $T_a$ bias offsets) has a
sizeable temporal variability that can be controlled by the monsoon weather conditions (such as
ambient air temperature, humidity and incoming radiation) and can sometimes be independent of
the glacier size (Figure 7). Whilst we cannot determine the impact of monsoon timing and
intensity upon the climatic sensitivity of these glaciers with the current dataset, we are able to
determine that the observed relationship to flowline distance is consistent to that of other regions
of the world (Figure 8). Future work on Tibetan glaciers should attempt to extend monitoring to
the pre-monsoon period to identify if a seasonal onset for the changing glacier climatic sensitivity
can be defined, and how the monsoon may affect it. Particular focus should be given to understand
the local meteorological conditions for each glacier, as this may explain some of the variability
in $T_a$ offset values, and why they may sometimes be independent of the along-flowline distance
(Figure 7).

466       *6.2.     Parameterising glacier climatic sensitivity*
In this study, we discuss the climatic sensitivity of on-glacier $T_a$ based upon observations above
a threshold ambient temperature for the onset of katabatic conditions ($T^*$). This sensitivity to
ambient temperature during relatively warm conditions, indicated by the $k2$ parameter of Shea
and Moore (2010)(Figure 1), demonstrates a generally consistent behaviour between the T-logger



observations of Parlung glaciers and those where this model had been previously implemented
(Shea and Moore, 2010; Carturan et al., 2015). It also resulted in a similar parameterisation, albeit
with a slightly greater sensitivity to the ambient temperature (i.e. larger $k2$ values - Figure 8b).
Whilst the newly presented dataset for the Parlung catchment provides an important confirmation
of the climatic sensitivity for some Tibetan glaciers, further studies of individual glaciers can
provide only local parameterisations for climatic sensitivity that may not be applicable to other
sites. Accordingly, we have made here a first attempt at combining many of the published datasets
regarding distributed $T_a$ on mountain glaciers around the world (Table 2) to examine the potential
for generalisability of a model accounting for climatic sensitivity (Figure 8).
We found a sizeable spread in the climatic sensitivities of $T_a$ for the on-glacier datasets considered,
though a consistently rapid decrease of sensitivity along glacier flowlines is found for most sites
up until ~2000-3000 m of distance (Figure 8b). While localised meteorological and topographic
factors likely interact to explain the spread of sensitivities at small flowline distances (Figure 8b),
the results suggest that small glaciers with flow lengths < 1000 m would reflect a 0.7-0.8
sensitivity to changes in $T_aAmb$. Beyond this distance, the climatic sensitivities notably follow
one of two patterns; a continued, albeit less rapid decrease in sensitivity (more generally following
the model proposed by Shea and Moore (2010)), or a tendency toward increasing sensitivity at
the largest flowline distances (more related to those findings of the 'ModGB' model - Figure 1a).
With reference to the relative $T_a$ differences among only on-glacier observations, these have been
termed as down-glacier 'cooling' or 'warming', respectively for many past studies (Ayala et al.,
2015; Carturan et al., 2015; Shaw et al., 2017; Troxler et al., 2020). Whilst the former is generally
associated with relatively warmer regions of study (Figure 9), such as the Canadian Rockies (Shea
and Moore, 2010) or Universidad Glacier (Bravo et al., 2017), no strong relationship of the
climate setting exists between these sites to explain the magnitude of the climatic sensitivity (i.e.
the strength of the glacier cooling and dampening effect) nor the observed increases in climatic
sensitivity on glacier termini (Ayala et al., 2015; Shaw et al., 2017; Troxler et al., 2020).
Interestingly, we noted that the most distant station observation used to derive the
parameterisation by Shea and Moore (2010) was located only around 60% of the total glacier
flowline distance (Bridge Glacier - Figure 10), whereas data presented by other studies, provided
observations up to the glacier terminus (Greuell and Böhm, 1998; Ayala et al., 2015; Shaw et al.,
2017; Troxler et al., 2020), therefore potentially parameterising different effects of the glacier
boundary layer. It has been suggested that observations at large flowline distances (such as that
on Parlung4 or Bridge Glacier) represent a segment of the boundary layer where the near-surface
layer becomes highly insensitive to the ambient free-air temperature fluctuations (point '3' in
Figure 1a and d). This phenomenon has been shown to be sustained over large fetch distances by
an increasing depth of the glacier wind layer (van den Broeke et al., 1997; Greuell and Böhm,
1998; Shea and Moore, 2010, Jiskoot and Mueller, 2012). However, as air parcels travel down-
glacier toward the glacier terminus (point '4' in Figure 1a and d), they potentially encounter warm
air entrainment due to a divergent boundary layer (Munro, 2006), up-valley winds (Pellicciotti et
al., 2008; Oerlemans, 2010; Petersen and Pellicciotti, 2011) or heating from debris-covered ice at
the terminus (Brock et al., 2010; Shaw et al., 2016; Steiner and Pellicciotti, 2016; Bonekamp et
al., 2020). These are effects of the glacier boundary layer that the ModGB model was designed
to account for, though we did not explicitly test this within our study due to a requirement for
more data and a greater number of parameters and assumptions (Shaw et al., 2017). The strength
of this so called along-glacier 'warming effect' could therefore be governed by local topography
(adjusting the boundary layer convergence or divergence) or the total glacier flowline distance
and the large fetch of a cool air parcel overcoming the competing effect of warm, up-valley winds
(Figure 1d - as seen at T2$_4$ in Figure 5).
By subjectively grouping glaciers by the presence of the observed increase in climatic sensitivity
and normalising the flowline distance of the observations by the total flowline for each glacier,
we identify that the relative increases in climatic sensitivity begin at ~ 70% of the total flowline
distance (Figure 10). A smaller climatic sensitivity can be observed for larger glaciers (Figure


10a), which is consistent with the development of the *KBL* over a large fetch (Greuell and Böhm,
1998; Shea and Moore, 2010), though the length itself indicates nothing clear about why greater
climatic sensitivity exists for some glacier termini (Figure 10b).
The clear outlier of these datasets is Juncal Norte Glacier in Chile (Figure 8b). It is interesting to
note that Juncal Norte is the only reported case in the literature on $T_a$ variability where the warmest
hours of the afternoon correspond to the dominance of an up-valley, off-glacier wind (Pellicciotti
et al., 2008; Petersen and Pellicciotti, 2011). Counter to the typical role of the dominant, down-
glacier wind layer for these warmest afternoon hours (Greuell et al., 1997; Greuell and Böhm,
1998; Strasser et al., 2004; Jiskoot and Mueller, 2012; Shaw et al., 2017; Troxler et al., 2020), up-
valley winds on Juncal Norte seemingly erode the along-flowline reduction in climatic sensitivity
(along-flowline cooling) up to a distance along the flowline where it is theoretically at its
maximum (point '3' in Figure 1). Evidence from other glaciers suggest that this point is close to
upper observations for Juncal Norte at ~70% of the total flowline (Figure 10b), though further
observations would be required to test this.
### 6.3. Future directions for researching air temperatures on glaciers
A limitation of our work is the dependency of the derived 'global' climatic sensitivities (Figure
8b) to the available off-glacier data and the published lapse rates to extrapolate them to the
relevant elevations on-glacier. In our case, we are able to identify a potentially non-linear lapse
rate of $T_aAmb$ for the highest elevations over Parlung94 and Parlung390 (Figure 4). Although we
cannot confirm this without off-glacier observations above the top of the flowline (Carturan et al.,
2015), we are able to well constrain ambient air temperature distribution using hourly
observations at several off-glacier locations to derive the best possible 'catchment lapse rate'. For
other datasets (Table 2), we rely upon the available off-glacier data and lapse rates that are not
derived in a consistent manner. The derivation of flowline distances from the DEM are also not
consistent between the prior studies (Shea and Moore, 2010; Carturan et al., 2015; Shaw et al.,
2017; Bravo et al., 2019; Troxler et al., 2020), and may hold some small influence on the derived
parameterisations (Table 3), particularly at lateral locations on the glacier (not explored here),
that can be subject to different micro-meteorological effects (van de Wal, 1992; Hannah et al.,
2000; Shaw et al., 2017). Equally, the uncertainty of the actual observations (e.g. section 3.2) is
hard to clearly define due the variable instrumentation (sensors and radiation shielding), on-
glacier location and local topographic and micro-meteorological effects of each study site (Table
2). Further work on a unified model of estimating $T_a$ should need to address these issues, perhaps
with further, dedicated analyses.
In our study, we apply the parameterisation of Carturan et al. (2015) to derive along-flowline
values of the theoretical onset of the *KBL* (*T\**). While these values appear appropriate for our case
studies (based upon manual inspection), they were derived for a more limited number of total
observations. We experimented with a static *T\** value of 5°C in order to test the sensitivity of our
analysis to the assumptions of *T\**, though found a minimal change in our derived *k2* sensitivities
(not shown).
Finally, in this study we assess climatic sensitivity based upon ambient air temperatures above
this *T\** threshold. We identify, however, that this is partly different to the climatic sensitivity
presented by earlier works (Greuell et al., 1997; Greuell and Böhm, 1998; Oerlemans, 2001;
2010), which considered all hours of the on-glacier observations when comparing to extrapolated
off-glacier $T_a$. In some instances, over estimation of on-glacier $T_a$ also for cooler conditions may
produce a consistent 'all-hour' climatic sensitivity value (i.e. where *k1* and *k2* sensitivities are
similar - Figure 1b). However, ignoring separate effects (*k1* and *k2*) due the rise of the *KBL*
(Figure 1c, Figure 5) arguably over-simplifies the glacier's climatic sensitivity and therefore does
not aptly describe the two behaviours separated by an onset event (Shea and Moore, 2010; Jiskoot
and Mueller, 2012). Accordingly, we caution somewhat the direct comparison of the climatic
sensitivity presented here and that of previous works, though consider the use of *k2* to be an
appropriate indicator of climatic sensitivity for this work going forward. As previously





mentioned, we have considered the approach of Shea and Moore (2010) to be a more
generalizable method for calculating glacier climatic sensitivity and thus estimating on-glacier $T_a$.
However, the competing effects of glacier katabatic and up-valley winds need to be incorporated
to address the challenges that less simplistic methods (i.e. ModGB) were designed for.
Based upon the findings of this work, we recommend that future research i) attempt to standardise,
where possible, the measurement and comparison of off- and on-glacier air temperature,
potentially exploring more the use of artificially-ventilated radiation shields that are less prone to
heating errors (Georges and Kaser, 2002), ii) instrument glaciers of varying size in the same
catchment to explore the relative importance of glacier size and local meteorological conditions
(Figure 7), and iii) model the detailed interactions of air flows on the glacier termini using, for
example, large eddy simulations (Sauter and Galos, 2016; Bonekamp et al., 2020) in order to
identify possible drivers of the observed increase in climatic sensitivity for certain glaciers (point
'4' in Figure 1).

## 7. Conclusions

We presented a new dataset of distributed on-glacier air temperatures for three glaciers of
different size in the south-east Tibetan Plateau during two summers (July - September). We
analysed the along-flowline air temperature distribution for all three glaciers and compared them
to the estimated ambient temperatures derived from several, local off-glacier stations. Using this
information, we parameterised the along-flowline climatic sensitivities of these glaciers using the
method proposed by Shea and Moore (2010) and presented the results in the context of several
available distributed on-glacier datasets to date. The key findings of this work are:

1. For our Tibetan case study, on-glacier air temperatures at short flowline distances are
   more climate sensitive (i.e. demonstrate a relationship with off-glacier air temperature
   that is closer to 1). We therefore confirm earlier evidence regarding the high sensitivity
   of small glaciers (flowline distances < 1000 m) to external climate, and thus future
   warming.
2. The largest offsets between observed on-glacier and estimated off-glacier air
   temperatures are found for the warmest off-glacier hours, during drier, clear sky
   conditions of the summer monsoon period.
3. Above the established onset of the katabatic boundary layer, climatic sensitivity to
   ambient temperature decreases rapidly up to ~2000-3000 m along the glacier flowline.
   Beyond this distance, both the Tibetan glaciers and other datasets of the literature show
   a slower decrease of climatic sensitivity.
4. A parameterisation for the climatic sensitivity of the Tibetan study glaciers implies a
   smaller boundary layer effect than the existing parameterisation of Shea and Moore
   (2010). The climatology of a given region may influence the magnitude of the glacier's
   climatic sensitivity, though no clear relationships with the climatology of the glacier sites
   are found, thus suggesting the stronger role of local meteorological or topographic effects
   on the along-flowline pattern of $T_a$ variability.
5. The terminus of some glaciers remain associated with other warm air processes, possibly
   due to boundary layer divergence, warm up-valley winds or debris cover heating. We find
   that these effects are evident only beyond ~70% of the total glacier flowline distance,
   although further work is required to explain this behaviour. A better understanding of
   temperature variability for this lower 30% is highly important as most of the summer
   melting will occur for this sector of the glacier.

In summarising the findings from all available distributed on-glacier datasets to date, we identify
some key directions for future work on this subject. This includes comparing local influences of
glacier size and micro-meteorology and standardising measurement practices, where possible, to
aid the conclusions for a generalised model of on-glacier air temperature estimation.



## Acknowledgements

Funding for the instrumentation of the Parlung catchment was provided by NSFC project (91647205 and 4191101270) and Newton Advanced Fellowship (NA170325). Á. Ayala acknowledges support from a FONDECYT project (number 3190732) and C. Bravo from the ANID Becas Chile PhD scholarship program. F. Pellicciotti acknowledges an ERC Consolidator Grant: 'RAVEN' (Rapid mass loss of debris covered glaciers in High Mountain Asia, grant agreement no. 772751). The authors kindly acknowledge the sharing of global datasets or parameters provided to aid this analysis, explicitly M. Nolan (McCall Glacier), J. Pomeroy, D. Pradhananga and the Global Water Futures Programme (Peyto Glacier), P. Smeets and IMAU, Utrecht (Pasterze Glacier), DGA, Chile (Universidad Glacier and Greve Glacier) and L. Carturan (AVDM, Italy). E. Ludewig is thanked for the provision of off-glacier temperature records at Sonnblick station, Austria. Additionally we recognise the hard work involved in obtaining and sharing all of the datasets acquired and the acknowledgements of those works.

## Author contributions

TES and WY discussed and designed the research plan with Parlung data provided by WY and CZ. Additional data and analysis was provided by AA and CB. TES wrote the manuscript with scientific input from all co-authors.

## Data availability

Calculated flowlines and climatic sensitivities are available at the following Zenodo repository: http://doi.org/10.5281/zenodo.3937777

## Competing Interests

The authors declare that they have no conflicting interests.

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

**Figures**





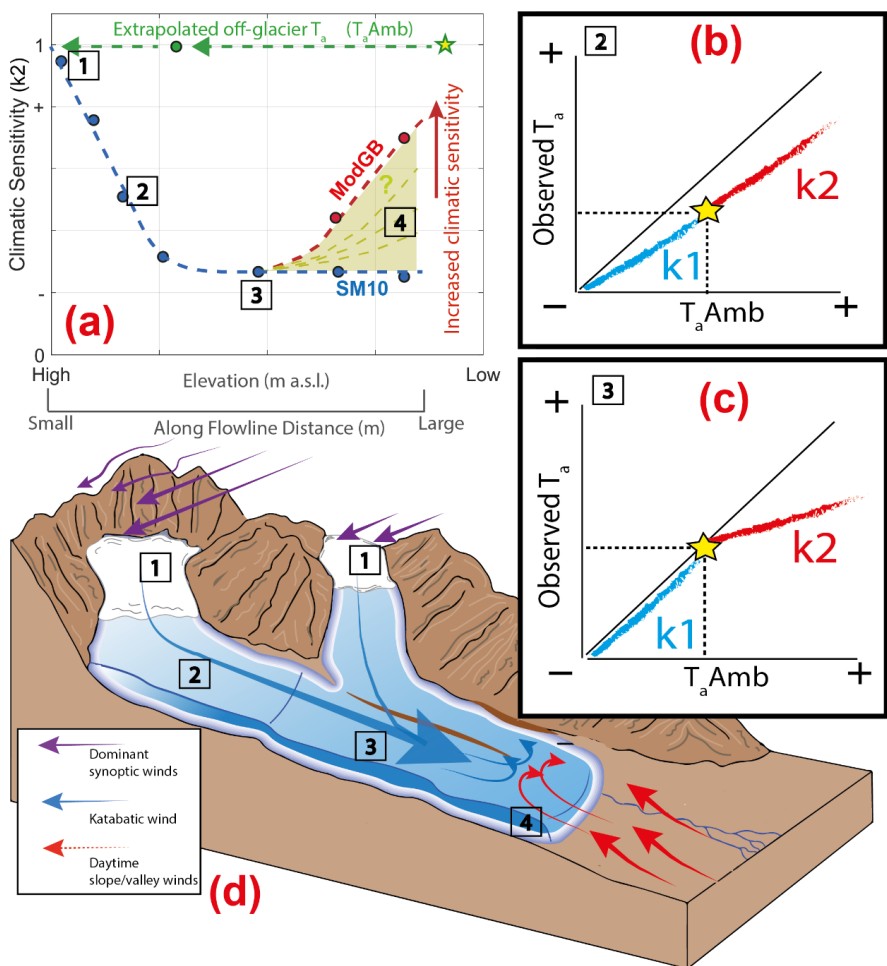

*Figure 1: A schematic diagram to describe the climatic sensitivity of on-glacier air temperature ($T_a$) to the*
*extrapolated ambient temperature ($T_a$amb) at given elevations/flowline distances on a mountain glacier.*
*Points 1-4 indicate locations of interest in the context of the current science for this topic that are linked*
*between panels. Panel (a) indicates the along-flowline 'k2' climatic sensitivities to $T_a$Amb, considering*
*down-glacier decrease in sensitivities and the observed differences in the models of SM10 and ModGB for*
*glacier termini (see text). While ModGB does not explicitly include the k2 parameter, its approach is similar*
*to considering an increasing climatic sensitivity to $T_a$Amb (see Ayala et al., 2015). The green line in panel*
*(a) indicates the local off-glacier lapse rate to estimate $T_a$Amb using off-glacier observations at varying*
*elevations (green dot). Panels (b) and (c) represent the differences of k1 and k2 sensitivities observed in*
*the data at different theoretical locations on the glacier (see Figure 5, for examples on Parlung glaciers),*
*the latter of which shows the theoretical parameterisation presented by Shea and Moore (2010). Panel (d)*
*represents an idealised case of katabatic and valley/synoptic wind interactions that potentially dictate the*
*along-flowline structure of on-glacier climatic sensitivity and thus $T_a$ estimation.*
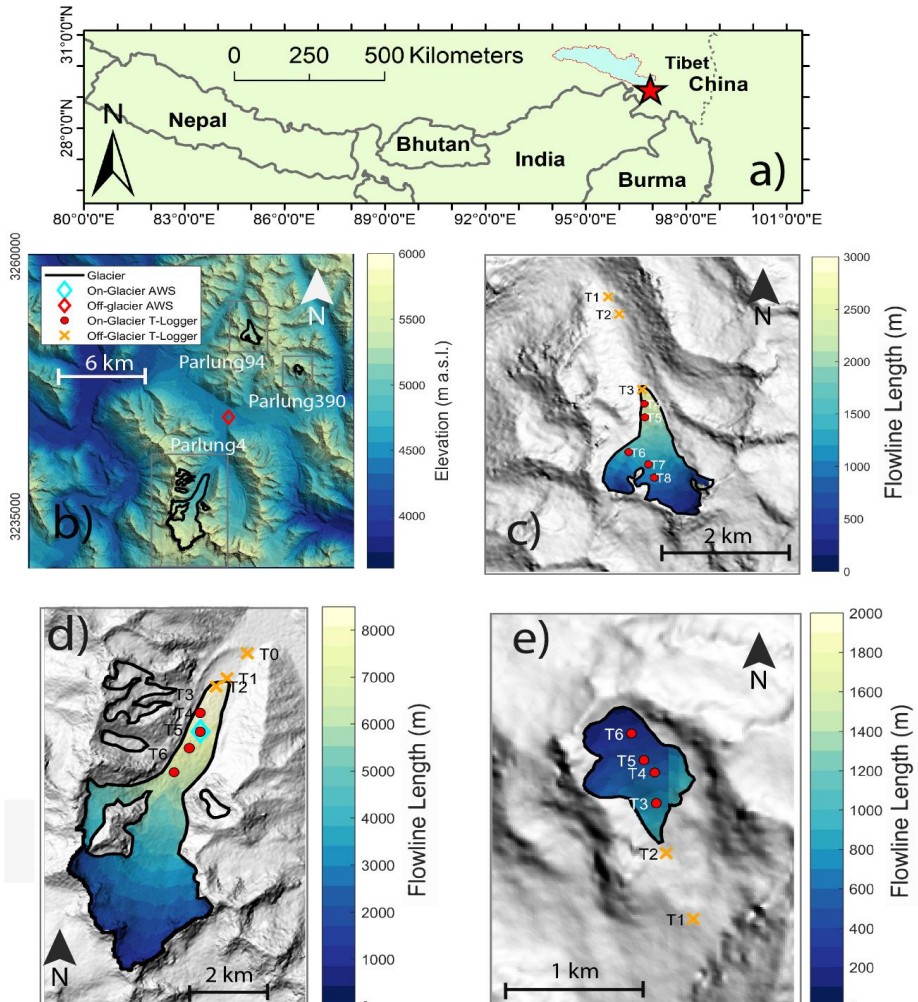

*Figure 2: The location of Parlung catchment in Tibet (a) and a map of the Parlung catchment (b) with the*
*study glaciers, Parlung 94 (c), Parlung4 (d) and Parlung390 (e). Off-glacier and on-glacier AWS and T-*
*Logger locations are shown (without glacier number suffix). (a) shows the elevation of the catchment (DEM*
*source: Alos Palsar) and (b-d) show the calculated flowline distances based upon TopoToolbox (scales*
*vary).*



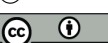


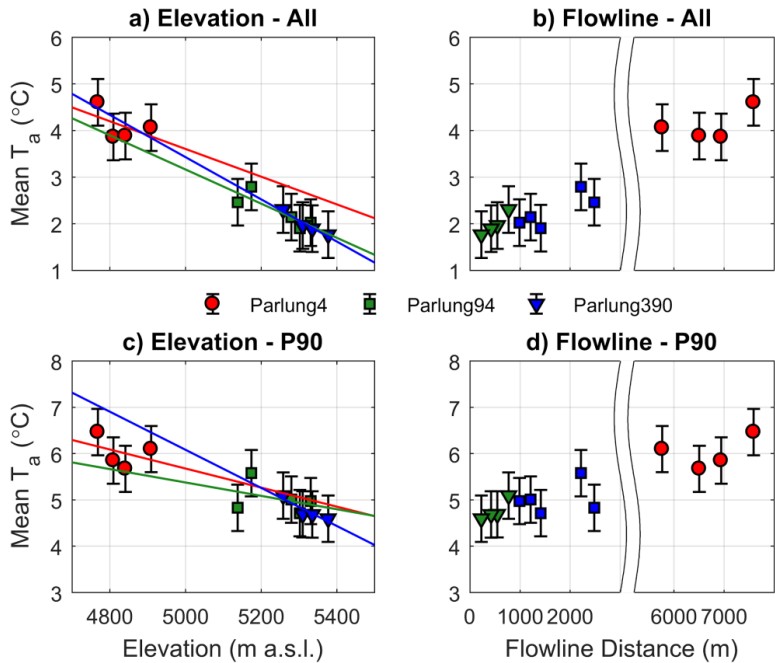

*Figure 3: The elevation-mean $T_a$ and uncertainty (errorbar) for (a) all hours and (c) P90 hours (n = 312).*
*Panels (b) and (d) are the equivalent plots against flowline distance. Coloured lines show the linear fit*
*against elevation ('lapse rate') to each glacier. An x-axis scale break is used in (b) and (d) for clarity.*



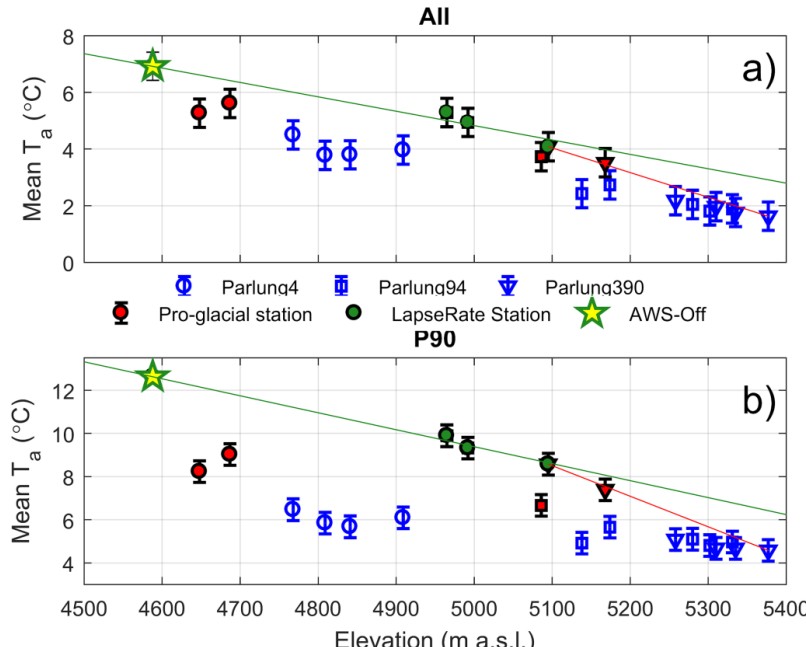


*Figure 4: The mean $T_a$ against elevation for all hours (a) and P90 hours (b), where blue markers are on-*
*glacier T-Loggers, red markers are pro-glacial T-Loggers and green circles denote off-glacier T-Loggers*
*used to construct an hourly variable 'catchment lapse rate' (green line), extrapolated from AWS_Off*
*(star). The red line indicates the piecewise lapse rate above the elevation of T1_390 to lapse $T_a$ to the top*
*of the flowline. A 0.5°C uncertainty is shown by the errorbar for each station (not applied to the lapse*
*rate for neatness).*





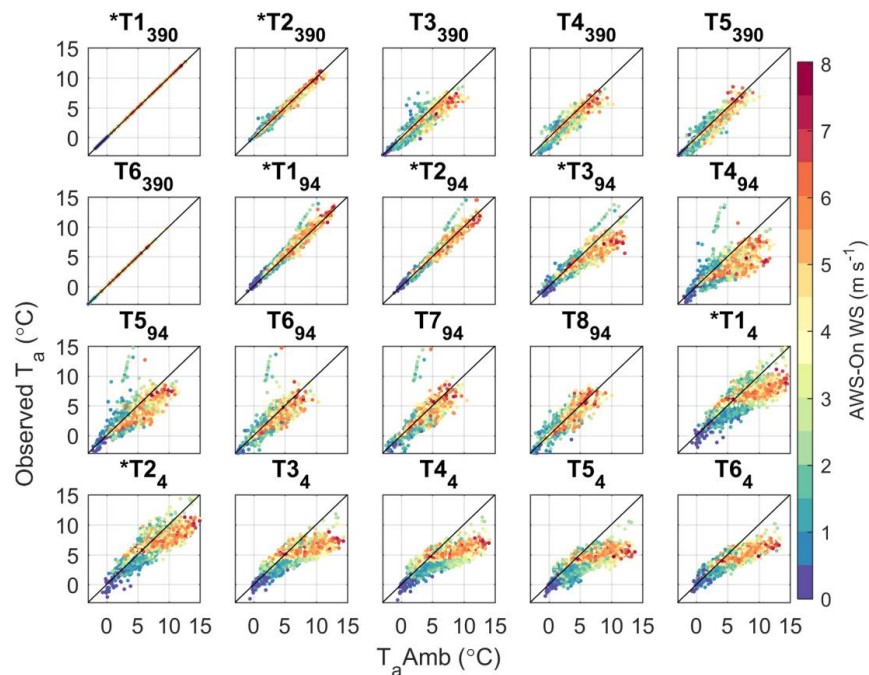

*Figure 5: Estimated ($T_a$Amb) vs Observed $T_a$ at each T-Logger location (including off-glacier T-*
*Loggers). Individual, hourly values are coloured by the observed wind speeds at AWS_On (Parlung4). No*
*on-glacier wind speed data exist for Parlung94 and Parlung390, so the coloured markers are only*
*assumed for those glaciers from the parlung4 wind speed data. * denotes stations that are off-glacier.*




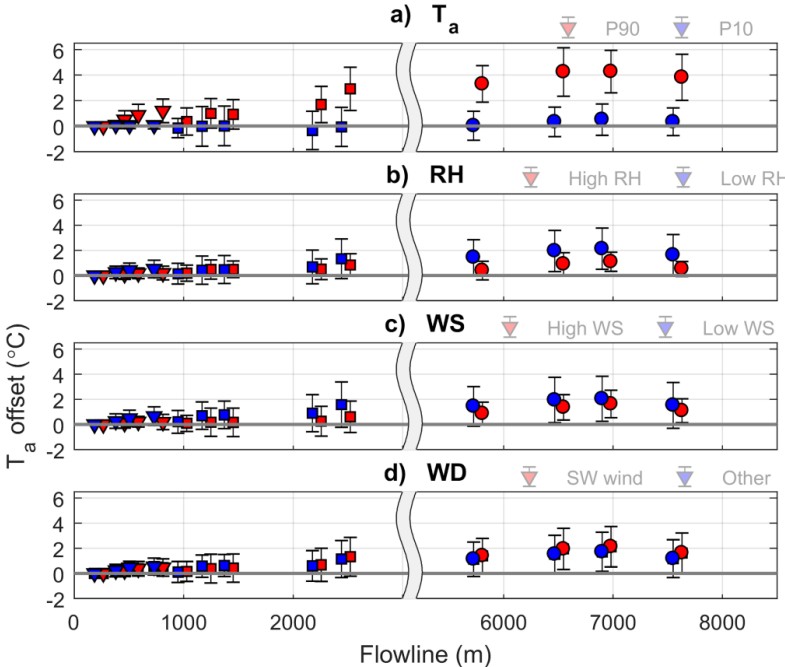

*Figure 6: The mean and standard deviation (error-bars) of hourly $T_a$ bias offsets (estimated-observed) along the glacier flowline. Each panel depicts hourly grouping by (a) off-glacier $T_a$ at AWS_Off (P90 is ≥ 10.5 °C and P10 is ≤ 3.5 °C), (b) off-glacier RH at AWS_Off (high is > 90 % and low is < 70 %), (c) wind speed from ERA5 (high = > 2.5 m s⁻¹ and low = < 0.7 m s⁻¹) and (d) dominant wind direction from ERA5 (Southwest wind direction is considered as 180-270°). Marker shapes show the different glaciers, as in Figure 3 and 4.*





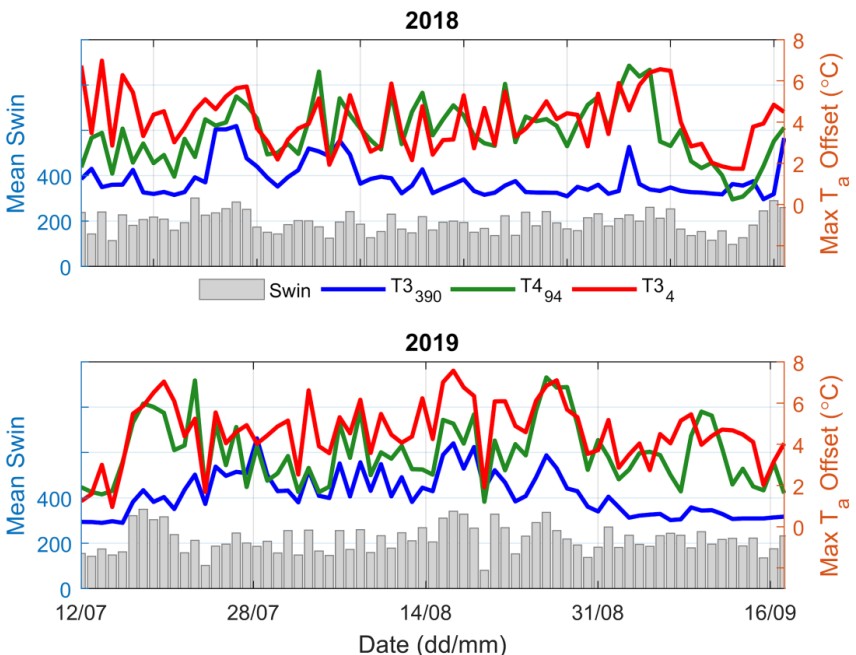


*Figure 7: Maximum daily $T_a$ offsets (estimated - observed) at the most distant along-flowline T-Loggers on each glacier for (top) and 2019 (bottom). Mean daily incoming shortwave radiation at AWS_Off is shown by the grey bars.*


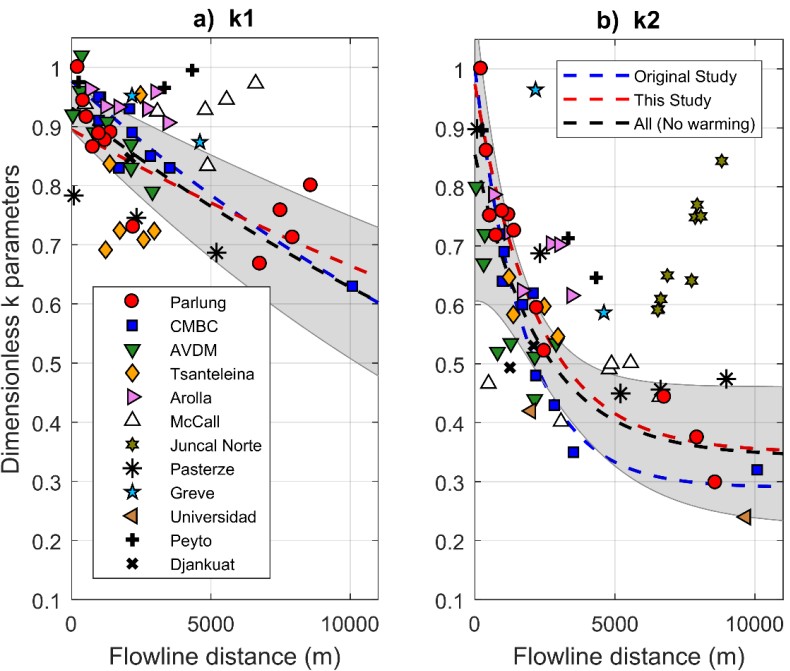


*Figure 8: The calculated k1 and k2 sensitivities as a function of the flowline distance of each observation on the Parlung glaciers (red circles) and other, global datasets (Table 2). The dashed blue and red lines show the fitted exponential parameterisation of Shea and Moore (2010) and this study, respectively. The dashed black line and shaded area denotes the equivalent parameterisation for all observations where a large increase in sensitivity on the glacier terminus ('warming effect') is absent (explicitly excluding data from McCall, Juncal Norte and Djankuat). The shaded area represents the 95% confidence interval of this fit line.*



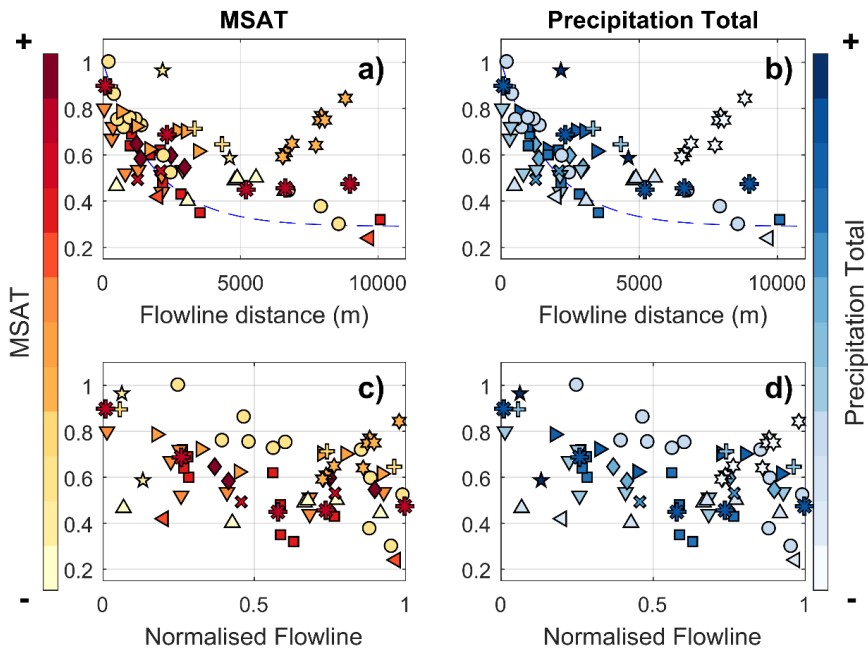

*Figure 9: The k2 sensitivities as a function of flowline distance (top) and a normalized distance, considering the total flowline distance for the year of study (bottom). The individual glaciers of grouped studies (Parlung, CMBC and AVDM) are separated and normalized by the individual glacier length (symbols as in Figure 8). Glaciers are coloured by rankings of the mean summer air temperatures (MSAT - left) and precipitation total (PT - right). The original parameterisation is retained in the top panels.*





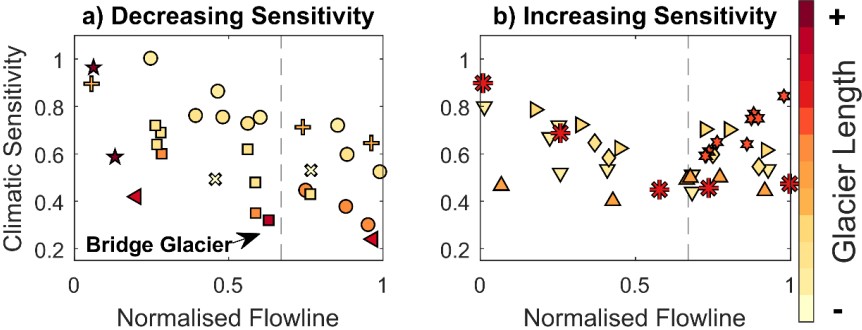


*Figure 10: The k2 sensitivity along the normalized flowline compared to total glacier length (colour bar)*
*and subjectively grouped by the evidence of a relative warming effect (increasing climatic sensitivity)*
*toward the glacier terminus.*












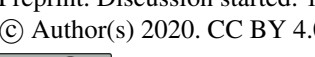



**Tables**

*Table 1: Details of each AWS/T-Logger station used in this analysis including the calculated flowline*
*distances.*

| Station | Latitude | Longtitude | Elevation (m a.s.l.) | Flowline (m) | on/off glacier |
|---|---|---|---|---|---|
| AWS_Off | 29.314 | 96.955 | 4588 | - | off |
| AWS_On | 29.500 | 97.009 | 4649 | - | off |
| $T1_{390}$ | 29.348 | 97.022 | 5095 | - | off |
| $T2_{390}$ | 29.352 | 97.020 | 5168 | - | off |
| $T3_{390}$ | 29.354 | 97.0202 | 5258 | 770 | on |
| $T4_{390}$ | 29.356 | 97.020 | 5310 | 544 | on |
| $T5_{390}$ | 29.357 | 97.019 | 5335 | 420 | on |
| $T6_{390}$ | 29.359 | 97.018 | 5377 | 224 | on |
| $T1_{94}$ | 29.621 | 97.218 | 4965 | - | off |
| $T2_{94}$ | 29.417 | 96.99 | 4992 | - | off |
| $T3_{94}$ | 29.635 | 96.975 | 5086 | - | off |
| $T4_{94}$ | 29.596 | 97.065 | 5138 | 2481 | on |
| $T5_{94}$ | 29.56 | 97.067 | 5174 | 2215 | on |
| $T6_{94}$ | 29.466 | 97.023 | 5302 | 1411 | on |
| $T7_{94}$ | 29.434 | 97.080 | 5280 | 1208 | on |
| $T8_{94}$ | 29.399 | 97.097 | 5331 | 988 | on |
| $T1_4$ | 29.271 | 96.968 | 4690 | - | off |
| $T2_4$ | 29.368 | 96.935 | 4769 | - | off |
| $T3_4$ | 29.298 | 97.168 | 4809 | 8589 | on |
| $T4_4$ | 29.298 | 97.168 | 4809 | 7940 | on |
| $T5_4$ | 29.496 | 97.126 | 4841 | 7505 | on |
| $T6_4$ | 29.403 | 97.068 | 4909 | 6765 | on |
















*Table 2: The details of each site where distributed on-glacier air temperatures are available. Elevation*
*ranges and mean summer air temperatures (MSAT) are reported for the year of investigation. Precipitation*
*totals (mm – 'PT') was obtained upon cited literature.*

| Site | Lat | Lon | Year(s) | Elevation | MSAT | PT | $T_a$ **Data Reference** |
|---|---|---|---|---|---|---|---|
| | | | | m .a.s.l. | °C | mm | |
| Parlung (Tibet) | 29.24 | 96.93 | 2018-2019 | 4600-5800 | 2.19 | 679 | This Study |
| CMBC (Canada) | 50.32 | -122.48 | 2006-2008 | 1375-2898 | 10.29 | 1113 | Shea and Moore (2010) |
| AVDM (Italy) | 46.42 | 10.62 | 2010-2011 | 2650-3769 | 7.94 | 784 | Carturan et al. (2015) |
| Tsanteleina (Italy) | 45.48 | 7.06 | 2015 | 2800-3445 | 13.76 | 805 | Shaw et al., (2017) |
| Arolla (Switzerland) | 45.97 | 7.52 | 2010 | 2550-3520 | 7.28 | 1663 | Ayala et al. (2015) |
| McCall (USA) | 69.31 | -143.85 | 2004-2014 | 1375-2365 | -2.28 | 500 | Troxler et al. (2020) |
| Juncal Norte (Chile) | -33.01 | -70.09 | 2007-2008 | 2900-5910 | 6.58 | 352 | Ayala et al. (2015) |
| Greve (Chile) | -48.88 | -73.52 | 2015-2016 | 0-2400 | -0.1 | 12000 | Bravo et al. (2019) |
| Pasterze (Austria) | 47.09 | 12.71 | 1994 | 2150-3465 | 12.66 | 2761 | Greuell and Böhm, (1998) |
| Universidad (Chile) | -34.69 | -70.33 | 2009-2010 | 2463-4543 | 8.24 | 474 | Bravo et al. (2017) |
| Peyto (Canada) | 51.66 | -116.55 | 2011 | 2260-3000 | 2.94 | 800 | Pradhananga et al. (2020)* |
| Djankuat (Russia) | 43.20 | 42.77 | 2017 | 3210-4000 | 12.13 | 950 | Rets et al. (2019) |

*\*paper not yet submitted*
*Table 3: The coefficients of the original SM10 model and those fit to the k1 and k2 sensitivities on the*
*Parlung glaciers and all glaciers where no warming effect was evident (see Figure 10).*

| Model | $k1 = β1*exp(β2*DF)$ | $k2 = β3 + β4*exp(-β5*DF)$ |
|---|---|---|
| CMBC (Shea and Moore, 2010) | *β1* = **0.977**<br>*β2* = **-4.4e-5** | *β3* = **0.29**<br>*β4* = **0.71**<br>*β5* = **5.6e-4** |
| Parlung | *β1* = **0.894** (0.805,0.983)<br>*β2* = **-2.972e-5** (-5.543e-5,-4.0e-6) | *β3* = **0.349** (0.241,0.456)<br>*β4* = **0.624** (0.492,0.757)<br>*β5* = **4.4e-3** (1.7e-4,7.2e-4) |
| All (no increased sensitivity on glacier terminus) | *β1* = **0.923** (0.886,0.96)<br>*β2* = **-3.375e-5** (-5.543e-5,-4.0e-6) | *β3* = **0.343** (0.225,0.46)<br>*β4* = **0.511** (0.38,0.642)<br>*β5* = **4.2e-3** (1.5e-4,6.9e-4) |





