# Peer review of "Distributed summer air temperatures across mountain glaciers: climatic sensitivity and glacier size"

_The Cryosphere, 2020_

## Referee Comment (RC1) · Anonymous Referee #1 · 27 Oct 2020

Shaw and others present an analysis of air temperatures observed over the surfaces of several mountain glaciers on the Tibetan Plateau. The study follows similar work conducted in western Canada (Shea and Moore, 2010), the Italian Alps (Carturan and others, 2015), and Chile (Ayala and others, 2015), which demonstrates that air temperatures observed over the surface of melting glaciers are predictably different from off-glacier 'ambient' temperatures that would be expected at the same elevation. Flow length appears to play a control on air temperatures, and the authors investigate their findings from the Parlung cathment in the context of a number of similar studies and datasets. Overall the paper is well-written, though the language could be simplified in many places to reduce the manuscript length and improve readability.

General comments: - As the paper is mostly focused on the datasets from the Tibetan Plateau, the title is bit misleading. Yes, there are some comparisons made with other datasets, but the title implies a global scale analysis.

- Sec. 4.3; The methods for analyzing the 'global' temperature datasets given in Table 2 are well-motivated, but questionable. An assumption of a constant 6.5C/km lapse rate will likely result in substantial errors in the estimated ambient air temperatures, which will then skew the values calculated for k1 and k2. I think the approach here needs to be improved before such comparisons can be made.

- The parameter k2 is termed "climatic sensitivity", but this is also a bit confusing. I would interpret climatic sensitivity as being related to changes in climate (e.g. mm w.e. /C). Specifically, k2 is unitless and it defines the relation between Ta\_glac and Ta\_amb during katabatic flows, so it really just quantifies the "cooling effect" a glacier has on the air mass immediately above it (1 = no cooling, 0.10 = strong cooling). My suggestion would be to use a different term, though I'm not sure "cooling effect" is the right one.

Specific comments: - L26: remove "several, "

- L26: see comments on "climate sensitivity"

- L30: "slower decrease of climate sensitivity" is tough to parse. How about "Beyond this distance...glacier datasets show little additional cooling effects."

- L31-32: It's not the observations that are sensitive, it's the glaciers themselves: "In general, small glaciers...have little cooling effect and are thus highly sensitive to changes..."

- L37: the "beyond" here seems to imply past or in front of the glacier. Suggest rephrasing.

- L47: suggest "... the use of linear temperature gradients, typically ... "

- L48-53: suggest shortening and simplifying: "A free-air ELR cannot be reliably used to
estimate near-surface air temperatures above melting glaciers, where steep gradients are found within 10 m of the surface (REFS)."

- L55: clarify that the overestimation occurs in energy balance models.

- L65-69: This is important, and I'd suggest moving it upwards as a standalone paragraph that also highlights that there are two main models for melt estimation.

- L73: "western Canada" or "southern Coast Mountains", not Canadian Rockies.

- L83-84: for clarity, remove "based upon. . .(1998)" – Gruell and Bohm is described in the next sentence

- L88: remove ",however,"

- L95: "The ModGB..."

- L103: "...though it does not explicitly account for physical processes that would reduce the glacier cooling effect at the glacier terminus."

- L106: see general comments above

- L113: can you stick with 'transferrability'? "generalizability" is a mouthful, and not commonly used in this field

- L113-L114: "As such, the transferability of near-surface air temperature models remain largely unknown. Analysis ..."

- L115-116: simplify: "In this study, we use new datasets of on-glacier air temperatures observed at three glaciers ...."

- L118: the word 'distributed' is probably unneeded

- Sec 3.3 can be moved up and included with the 3.1, which can be renamed "Meteorological observations", but just start with "Air temperatures (Ta), incoming shortwave,..."

- L193-194: remove the first part: "Flowline distances (m) for each glacier were calcu-

TCD
lated with the TopoToolbox..."

- L200: remove "between the aforementioned studies."

- L206-207: Feels like this needs a more general introduction: "Our methods consist of (1) aggregating temperature observations based on off-glacier temperatures, (2) generating off-glacier temperature gradients to compare on and off-glacier temperatures, and (3) estimating the glacier cooling effect on near-surface temperatures by fitting parameters to the SM10 model." Your rationale can then be moved into the individual subsections.

- L226: replace "within the glacier boundary layer" with "between ambient and onglacier temperatures"

- L226 – 229: this could be simplified to read "We calculate hourly ambient lapse rates from the off-glacier weather station and temperature loggers T1\_94, T2\_94, and T1\_390. These sites are assumed to be unaffected by the glacier cooling effect."

- L234: "...T-logger to quantify the difference between ambient and on-glacier Ta at each site..."

- Eq 1: asterisks should be superscript in this equation, and see general comments above

- L276: suggest using 'transferability' instead of 'generalisability'
- L284: is the ELR suitable for all sites and seasons?
- L308: since there is a scale break in the figure, it is tough to see this as 'linear'

- L349 – 351: Simplify! "For P90 conditions (Fig 6a), differences between TaAMB and observed on-glacier temperatures are up to 5.8 C at flowline distances greater than 7000 m."

- L351: avoid 'heighten' in this context
- Sec. 5.2 : avoid using "offset"

- Figure 7: perhaps show these relations as scatterplots. Tough to interpret the relation using line and bar plots.

- L352 – 353: RH will typically be high when temperatures are low, so this is not surprising.

- L382: "notably distinct" doesn't really fit with the results in the table, as the coefficients of SM10 are within the confidence intervals of the other datasets

- L430: need to specify energy balance models. Temperature indexed approaches don't need this correction if the melt factors are calculated with off-glacier temperatures.

- L474: "While the data from the Parlung catchment..."

- L493: southern Coast Mountains (not Canadian Rockies)

---

## Referee Comment (RC2) · Luca Carturan (Referee) · 14 Nov 2020

**General comments**

The paper from Shaw et al. presents a new interesting dataset of on-glacier air temperature measurements, which are compared to local off-glacier measurements in order to estimate the climatic sensitivity along the glaciers, and its spatial and temporal variability. In addition, the authors compare their results with similar datasets collected around the world, making assessments on the generalizability of the Shea and Moore (2010) approach.

The new dataset is of relevance, also because it provides additional observations for long flowline distances, which have been undersampled so far. The results are interesting and substantially agree with former parameterizations, confirming their good generalizability. This is a nice intercomparison among different sites in different geographic areas, even if it is not the first as suggested the Introduction.

In my opinion the paper is rather well written, even if there are parts that require rewriting to be clear. I refer the authors to the specific comments, but I also suggest a careful proof reading (sorry I am not a native speaker).

One of my major concerns is related to the 'accuracy' assessment of the air temperature measurements. I have written a specific comment on this regard. In general, the authors compare measurement datasets collected in the last decades using rather different experimental setups, in particular for radiation shields that range from aspirated research-grade ones, to simple passive-ventilation tubes opened at both ends. The authors are well aware of this issue, as they write in the discussion and the outlook. I am not criticizing the single experimental designs and approaches, but I agree with them that this is one of the main issue for intercomparisons, often overlooked, and that in the future this aspect requires higher attention.

According to me, the authors should add some discussions on the influence of local glacier surface topography, and in particular the width/length ratio of glacier tongues (i.e. the heating from surrounding slopes that are free from ice cover) and the role of surface steepness upslope of Ta sites. Clearly, the Shea and Moore (2010) does not account for glacier slope, which however is important in regulating the local prevalence of the cooling caused by the loss of sensible heat over the adiabatic heating for descending air. It could be interesting to observe that the Greuell and Böhm (1998) approach accounts for this.

**Specific comments**

Title: please consider this or similar alternatives (less generic): Distributed summer air temperatures across three glaciers in the Tibetan Plateau: climatic sensitivity and comparison with existing glacier datasets

30: both the Ta 'on' (or over) the Tibetan glaciers

31: In general, observations on small glaciers (*with flowline shorter than 1000 m*) *reveal that they* are highly sensitive to temperature changes outside the glacier boundary layer. Or maybe I have misunderstood the meaning, a possible alternative could be: In general, *air temperature* on small glaciers (*with flowline shorter than 1000 m*) *is* highly sensitive to temperature changes outside the glacier boundary layer.

35: please rephrase, e.g. replace 'remain associated with other warm air' with 'i*s affected by warming*' processess that increase…

40: is still required to explain '*the variability*' of these effects '*on*' different glaciers

48-49: that or which?

51: maybe better 'increases' rather than 'heightens'?

74-76: please rephrase to improve readability of this sentence (and in general of the paper, adding some commas, or splitting long sentences)

77-78: have been tested by …..

82-86: this period requires commas and/or splitting

88: adds, additional, is repetition ('additional' should be removed in my opinion, also because there is another one at the end of the period)

95: consider replacing 'Thus' with 'In this way'. The separation from the period above is questionable in my opinion

98: Because distributed on-glacier observations are often limited, and rarely cover the full length of the glacier boundary layer (or flowline?), this additional correction for warming associated to the unknown parameters (am I right?) of ModGB can lead to high uncertainty in Ta estimates at the glacier terminus (or maybe better 'lower part'?)

104: to be the cause of 'the' relative warming 'at' the glacier terminus

107-109: in my opinion, the simplicity and the statistical nature of an approach are not sufficient by themselves to make it more generalizable than more complex schemes. Maybe you meant 'easily applicable'? The generalizability of the Shea and Moore (2010) approach was instead assessed by previous works and applications to other glaciers. Please consider revising.

119: on-glacier observations available around the world. (I would delete 'made to date').

111-112: I do not agree with this statement. Please see for example the comparisons made by Greuell and Böhm (1998) in their Figure 5, and by Carturan et al., (2015) in their figures 6 and 8

130: 'wider Parlung catchment' is not so clear. Please see also my comments in this regard for Figure 2.

137: and the 'longest' rain season? (I would remove 'annual')

141: please clarify what is the 'more elevation-independent mass balance sensitivity' (lower sensitivity of mass balance to elevation?)

142: 'Because Tibetan glaciers are shrinking and fragmenting, the accurate estimation of on-glacier temperatures is relevant for investigating and modelling their climatic sensitivity (Carturan et al., 2015). However, to date, no studies regarding the distribution of on-glacier temperature have been performed within the Tibetan Plateau'

149: in this section you report the manufacturer and model of temperature sensors but information on manufacturer and model of radiation shields is of equal (or even larger) importance. Please add this information.

156: from 12th July 'to' 18th September (exactly the same day/month in both years?)

157: here you mention Table 1 after speaking about data gaps, but Table 1 does not report this information, please check or modify

163-164: the sentence from 'For' to 'station' leaves the reader with incomplete information on the spatialization of off-glacier air temperature. Is it in the following sections? If so, move this sentence there; if not, complete here

169: in my opinion this intercomparison is useful, but it does not reveal the 'uncertainty' of measurements, which would require an intercomparison among instruments and shields under controlled conditions (see

e.g.: https://www.wmo.int/pages/prog/www/IMOP/PastIntercomparisons.html). Instead, it enables to assess the 'comparability' of measurements, which is relevant in this work but is different from uncertainty (it would require at least a comparison towards an artificially-ventilated radiation shield). I want to highlight this difference because both $T4_4$ and AWS_On have passive-ventilation systems, which are subject to the same error under similar conditions. For example, they are likely to lead to significant overestimation of air temperature under high radiation and low wind speed. In these circumstances a small difference between the two instruments does not reveal that they are affected by small uncertainty, it only enables to state that they are comparable. I have measured differences up to several degrees in these conditions between passive- and active-ventilation systems on glaciers, especially with high surface albedo. This is the reason why I prefer using active-ventilation systems, even if I perfectly understand the advantage of using passive ventilation systems, which in harsh high-mountain environments are often the only option (also considering costs, power requirement and possibility of failure of active systems).

170: in table 1, $T4_4$ is at 4809 m (same elevation of $T3_4$), whereas AWS_On is at 4649 m. On the other hand, in Figure 2$T4_4$ and AWS_On are at the same place, therefore I think there is an error in Table 1

171: that are located (remove 'co-')

174: this 'P90' is not clear, it looks like an alternative name for the AWS_Off. Please clarify, for example you could write: '……temperature at AWS_Off. We find that for these warm hours (hereafter referred to as 'P90' - (Ayala et al., 2015; Shaw et al., 2017; Troxler et al., 2020)), when the KBL development is theoretically at its strongest (e.g. van den Broeke, 1997; Oerlemans and Grisogono, 2002), 95% of hourly differences……'

183: based on these results, did you consider filtering the temperature data, e.g. excluding those with ERA5 wind speed lower that 2 m/s?

186-188: 'In addition to Ta, in this work we used incoming shortwave radiation and relative humidity measured at AWS_Off, on-glacier wind speed measured at AWS_On, and 'free-air' wind speed and direction derived from ERA5 reanalysis data (C3S, 2017)'.

192: to obtain elevation information

201: please remove 'that' before 'the impact'

202: at few 'selected' points

206: 'for aggregation of on-glacier ta data into sub-groups, and for calculating the distribution of off-glacier ta in space.

209: and 'a quantification of' the effect of the glacier bl. (is referred to the 'allows' above)

212: are 'described' in sections…

214: please state explicitly to what P10 and P90 correspond, i.e. the 10% coldest and the 10% warmest off-glacier temperature? and what about data between them?

217-219: unclear

219-222: can be written more clearly, e.g. add 'on-glacier' to Ta (be explicit)

234: consider removing 'bias'

278-279: I do not understand this sentence from 'we' to 'site' (maybe a repetition of what is reported at L286-287?)

299: what is 'the best available value from the relevant literature' for the annual precipitation at the AVDM site? Carturan et al., (2015) report 1233 mean annual precipitation at that site from 1979 to 2009, therefore I wonder where the authors found 784 mm as reported in Table 2 (it looks like uncorrected raw data from the Careser diga weather station….). In 2010 and 2011 the annual precipitation was above average (i.e. about 1400 mm).

308: please report n for all hours

309: flowline 'distance' (or DF), also in the following

312: and 'with' flowline distance

321: when comparing 'Ta' to glacier elevation or flowline (distance)?

324: please add 'at the same elevation' after 'Ta'

335: 'i.e. where on-glacier observations are expected to match Taamb'

340-342: I understand the meaning but this sentence should be written more clearly, e.g. there is a good alignment between on/off-glacier temperature until the katabatic winds start to blow on Parlung4

345: and 'extension' of the katabatic wind into the proglacial area

347: also here consider removing 'bias' (in any case I would use bias or offset, but not both). In the following you use only offset so I would stick to that

351: These effects appear to 'increase' beyond 2000 m 352 along the flowline on the Parlung94, though 'significant' offsets are observed for all glaciers (and sites?)

360: would it be possible to state that, besides individual meteorological variables, the offset is largest for warm/anticyclonic conditions, and lowest for cool/cyclonic conditions?

364: the terminus 'of' each glacier

366: recorded at AWS_off

369: though 'it' varies….

371: '…September, while the offset for Parlung4 and Parlung94 remain significant'. Possible explanation for that?

376-387: this part is written rather poorly, please improve it (concepts are more clear and better described e.g. in the caption of Figure 8). In particular, try to avoid writing 'parameterisations' (too general) and write explicitly whether you are referring to 'sensitivities' or 'fitted exponential functions'

388-396: these results are interesting, and I wonder if there can be some influence from the very low width/length ratio of the tongues of McCall and Juncal Norte, surrounded by ice-free slopes that are expected to warm considerably… Parlung4 instead has a more 'compact' shape (i.e. less thin and long tongue). In addition I ask the authors to consider possible differences in the sensors and radiation shields used in the various referenced works and datasets (e.g. the radiation shields used in Juncal Norte were PVC cylinders opened at both ends, that are very different from the generally-used multi-plate radiation shieds)

399: the investigated sites lie close to the original SM10 exponential function up to ~4000 m….

416-423: poorly written, please clarify (requires multiple reading)

427: bring additional evidence of the spatial…

429: and highlights the need to appropriately account for these effects in glacier melt models

431: remove 'since'?

440: To the authors' knowledge?

487: remove 'more' before 'generally following…'

489: 'in agreement with the 'ModGB' model…'

499: the most distant station: the station with the largest flowline distance?

564: for a more limited number of total observations: do you mean 'from a smaller sample size of temperature observations'?

566: however we found negligible sensitivity of derived K2 on T* (If I understand the meaning correctly)

575: rise or onset or increasing strength? I would rewrite, e.g.: However, ignoring the differences in climatic sensitivity that have been reported before and after the onset of the KBL, is arguably an over-simplification and does not enable to describe correctly the observed behaviour

578-584: poorly written

588: please remove 'more' after exploring. Carturan et al. (2015) used these ventilated devices, are there other studies with ventilated shields among those compared in this work? I would add here the related references

591: 'at' the glacier terminus

593: for certain glaciers or glacier areas?

606: please replace 'are more climate sensitive' with 'display high climatic sensitivity'

607: that is close to 1 (remove 'r' at the end of closer, otherwise you should report the term of comparison)

618: compared to the existing parameterization

623: is associated with (not 'remain')

624: I would add (also in the discussion) the thermal (infrared) emission from the surrounding bare rocks, especially for thin and long ablation tongues

627: this is not true if the tongue is thin (small % area compared to the total glacier area). I would rewrite: is highly important because this is the glacier part that is most affected by ablation

633: to enable the construction of a generalised model for on-glacier air temperature estimation.

Figure 1: in my opinion the green parts of fig. 1a can be removed. I would also change the colour of (a), (b) and (c) from red to black, and try to uniform their placement (e.g. upper right corner of their frames). I would also shorten (and make more easily readable) the caption. The difference between (b) and (c) is not well explained. In a general scheme like this, I think that one should also find a short explanation of what k1, k2 and the yellow star represent

Figure 2: in b) I cannot see the map of a catchment, i.e. one expects to see the boundary of a hydrological catchment, with its divide, rivers and outlet. Please rephrase. When you mention suffix please refer to table 1. Please note that (A) does not show elevation data (it is (b)). c-e show the calculated flowline distances. Please consider homogenizing the scales for flowline distances. Off-glacier T2 on Parlung4 and T3 on Parlung94 actually look like 'on-glacier' in the figure.

Figure 3: caption 'The mean Ta against elevation…' (see caption Figure 4)

Figure 4: title of second chart (P90) is partially cut. Caption: 'The red line indicates the piecewise lapse rate above the elevation of T1_390  to the top of the flowline'

Figure 5: please write explicitly 'wind speed' in the y axis (WS is the acronym but generates confusion with the WS of 'AWS' to a quick reader). Caption: you can remove the sentence from 'No' to 'data', it is sufficient to describe it in the maintext.

Figure 6: why are the legends semi-transparent? Caption: (estimated 'from Taamb' - observed). high 'is' > 2.5 m s-1 and low 'is' < 0.7 m s-1)

Figure 7: in the caption: Maximum daily Ta offsets (estimated 'from Taamb' - observed) at the on-glacier T-Logger closest to the terminus of each glacier for 2018 (missing!) (top) and 2019 (bottom).

Figure 9: Caption: The original SM10 parameterisation is retained in the top panels

Figure 10: The k2 sensitivity along the normalized flowline compared to the total glacier length (colour bar). Glaciers have been grouped in two clusters: a) those with downglacier decreasing sensitivity, and b) those with increasing sensitivity towards the glacier terminus.

Figures 9 and 10: I find difficult to refer to Figure 8 for site coding (consider repeating the legend with geometric symbols in these figures)

Table 2: please check the precipitation data (e.g. AVDM and the 12000 mm in Chile). At line 981: 'Annual' precipitation totals….. Moreover, to which elevation are referred the MSAT and PT? I think it is relevant considering the high vertical lapse rates for both variables.

984: and those 'fitted'

---

## Author Comment (AC1) · 4 Dec 2020

Shaw and others present an analysis of air temperatures observed over the surfaces of several mountain glaciers on the Tibetan Plateau. The study follows similar work conducted in western Canada (Shea and Moore, 2010), the Italian Alps (Carturan and others, 2015), and Chile (Ayala and others, 2015), which demonstrates that air temperatures observed over the surface of melting glaciers are predictably different from off-glacier 'ambient' temperatures that would be expected at the same elevation. Flow length appears to play a control on air temperatures, and the authors investigate their findings from the Parlung catchment in the context of a number of similar studies and datasets. Overall the paper is well-written, though the language could be simplified in many places to reduce the manuscript length and improve readability.

*We thank the reviewer very much for their constructive comments on our manuscript. We have taken on board all their comments, and as a result we now have an improved manuscript. The main changes in the manuscript, in response to the reviewer's suggestions, are the following (which we respond to specifically below):*

1) *We have adjusted the title to more appropriately reflect the content of our work, giving specific mention to the Tibetan Plateau case study and given the revised terminology.*
2) *Simplified the text following both reviewer's comments to be more clear and and reduce the manuscript length in places.*
3) *Added additional analysis and discussion that clarifies the expected uncertainties in parameters of the Shea and Moore (2010) approach (i.e. k1 and k2 parameters) when using different lapse rates for off-glacier air temperature extrapolation at three of the 'global' sites.*
4) *Adjusted the terminology of climatic sensitivity to a temperature sensitivity throughout the manuscript.*
5) *Changed and adjusted a few figures based upon the requests of both reviewers (Figures 1, 3, 7 and 10).*

General comments: - As the paper is mostly focused on the datasets from the Tibetan Plateau, the title is bit misleading. Yes, there are some comparisons made with other datasets, but the title implies a global scale analysis.

*We thank the reviewer for raising this point. We have adjusted the title to: "Distributed summer air temperatures across mountain glaciers in the south-east Tibetan Plateau: temperature sensitivity and comparison with existing glacier datasets ".*

- Sec. 4.3; The methods for analyzing the 'global' temperature datasets given in Table 2 are well-motivated, but questionable. An assumption of a constant 6.5C/km lapse rate will likely result in substantial errors in the estimated ambient air temperatures, which will then skew the values calculated for k1 and k2. I think the approach here needs to be improved before such comparisons can be made.

*We thank the reviewer for their comment. We agree that using the environmental lapse rate (ELR) to calculate the sensitivity of the k1 and k2 parameters entail limitations (as the reviewer points out, the estimated air temperature might be affected by errors and so might be the k1 and k2 values). We note, however, that only three sites (Peyto, Universidad and Djankaut Glaciers) of the 11 examined relied upon the ELR to extrapolate air temperatures to the elevation of the on-glacier observation stations. For all remaining sites we used published values of the calculated lapse rates and/or local station data to construct our best possible 'catchment lapse rate' (i.e. for Parlung Glaciers).*

*To address the reviewer's comment for those three glaciers, where we relied upon the ELR in the absence of local lapse rates, we recalculated k1 and k2 with new lapse rates within a suitable range for off-glacier conditions in summer. We adjusted the lapse rate value +/- 1.5°C km-1 around the ELR (resulting in a range of -5 / -8°C km-1) and found changes of k2 within ~0.03 (see Figures R1- R3 below).*

*The differences in sensitivities (k2 parameter) for all three glaciers are in the range of <0.01 - 0.03, and are shown in the figures below, which correspond to Figure 8 of the original manuscript. It is apparent that the changes in the lapse rate result in negligible changes in both k2 as well as its relationship to the flowline.*

*The differences in the k1 parameters at all glaciers are <0.01 for the different tested lapse rates.*

*In addition to the test described above, for Peyto glacier we also calculated and tested an additional lapse rate, as on this glacier the uppermost on-glacier AWS is close to a mountain ridge at the upper limits of the glacier accumulation zone. Because of the short flowline distance at this location, we can assume that the boundary layer effect is minimal and temperatures on the glacier at this AWS approximately equal the 'ambient' off-glacier air temperature. We thus also tested an off-glacier to on-glacier lapse rate calculated between this on-glacier station and the off-glacier AWS (labelled 'Off-On' in Figure R1). This is important because higher elevations will often be more sensitive to changes in the temperature lapse rate (when temperature is extrapolated from a low elevation AWS). Using this 'Off-On' lapse rate, gives us a k2 value of 1 at the highest station, but provides an estimate of k2 at the other on-glacier AWS. We show in Figure R1 that these changes have a minimal impact on the k2 sensitivity on Peyto Glacier (maximum k2 difference = 0.02).*

*We have now indicated in the revised paper the glaciers where the ELR was used in Table 2 and have added a brief mention of the calculated k2 differences found from adjusting the lapse rate in the methodology section 4.4.*

[Figure]

*Figure R1: The calculated k2 parameters for Peyto Glacier using difference lapse rates. The lapse rates used are: ELR (-6.5°C km-1 - black), -5°C km-1 (blue), -8°C km-1 (red) and a lapse rate derived between the off-glacier forcing AWS and the highest elevation on-glacier AWS. For the latter, we assumed no boundary layer cooling of the glacier at the top of the flowline.*

[Figure]

*Figure R2: As Figure R1, but for Djankuat Glacier.*

[Figure]

*Figure R3: As Figure R1, but for Universidad Glacier.*

- The parameter k2 is termed "climatic sensitivity", but this is also a bit confusing. I would interpret climatic sensitivity as being related to changes in climate (e.g. mm w.e. /C). Specifically, k2 is unitless and it defines the relation between Ta_glac and Ta_amb during katabatic flows, so it really just quantifies the "cooling effect" a glacier has on the air mass immediately above it (1 = no cooling, 0.10 = strong cooling). My suggestion would be to use a different term, though I'm not sure "cooling effect" is the right one.

*We agree with the reviewer that "climatic sensitivity" is not an unambiguous term and is used with different*

*meanings by different scientific communities, and that our use here might thus appear slightly misleading to some readers. We use this term based on many of the previous works on air temperature estimation (Greuell and Böhm, 1998; Oerlemans, 2010; Carturan et al., 2015). It was first introduced by Greuell and Böhm (1998) to indicate the sensitivity of the glacier temperature to the ambient conditions. We would argue that the term 'cooling effect' is equally misleading due to the fact that temperatures are not only cooled (a static bias of X °C compared to the off-glacier temperature), but also that the diurnal cycle is 'dampened' (Carturan et al., 2015). Because the k1 and k2 parameters represent a ratio of the on and off-glacier temperature, they represent a sensitivity of the above-glacier temperature to the external conditions. We recognise that a climatic sensitivity is often used to indicate a rate of melting per unit temperature increase. We believe that the term temperature sensitivity is perhaps more appropriate in this study. To dispel ambiguities as much as possible, however, we have clearly explained our definition at the beginning of the manuscript, and also make now the distinction to the other meaning climate sensitivity as used in the literature. We have also made efforts to adjust this within the manuscript and provide specific definitions in the introduction to avoid any misinterpretation.*

Specific comments: - L26: remove "several, "

*We have adjusted this in line with the reviewers comments.*

- L26: see comments on "climate sensitivity"

*This has now been adjusted throughout, except where we give mention to the term 'climatic sensitivity' as termed by those earlier studies (e.g. Greuell and Böhm, 1998, Oerlemans, 2010).*

- L30: "slower decrease of climate sensitivity" is tough to parse. How about "Beyond this distance. . .glacier datasets show little additional cooling effects."

*We thank the reviewer for the suggestion and have adjusted this based upon their suggestion.*

- L31-32: It's not the observations that are sensitive, it's the glaciers themselves: "In general, small glaciers. . .have little cooling effect and are thus highly sensitive to changes. . ."

*Changed now based upon the reviewer's suggestion.*

- L37: the "beyond" here seems to imply past or in front of the glacier. Suggest rephrasing.

*We have now written as 'beyond this distance', referring to the 2000-3000 m flowline distance from the previous sentence.*

- L47: suggest ". . .the use of linear temperature gradients, typically..." - L48-53: suggest shortening and simplifying:

"A free-air ELR cannot be reliably used to estimate near-surface air temperatures above melting glaciers, where

steep gradients are found within 10 m of the surface (REFS)."

*We have adjusted this in line with the reviewers comments.*

- L55: clarify that the overestimation occurs in energy balance models.

*Done - we have added also that models of intermediate complexity (ETI models) are affected:*

*"While models applying the degree day approach can make use of off-glacier temperatures as forcing because they are heavily reliant on calibration, for physically based models and models of intermediate complexity (Pellicciotti et al., 2005; Ragettli et al., 2016) it is key to resolve the air temperature distribution over glaciers, especially for turbulent flux calculations."*

- L65-69: This is important, and I'd suggest moving it upwards as a standalone para graph that also highlights that there are two main models for melt estimation.

*We agree with the reviewer and have moved this earlier in the paragraph.*

- L73: "western Canada" or "southern Coast Mountains", not Canadian Rockies.

*Changed*

- L83-84: for clarity, remove "based upon. . .(1998)" – Gruell and Bohm is described in the next sentence

*Updated now.*

- L88: remove ",however,"

*Done*

- L95: "The ModGB. . ."

*We have adjusted this now*

- L103: ". . .though it does not explicitly account for physical processes that would reduce the glacier cooling effect at the glacier terminus."

*Updated with the reviewer's suggestion.*

- L106: see general comments above

*Changed now based upon the reviewer's suggestion.*

- L113: can you stick with 'transferrability'? "generalizability" is a mouthful, and not commonly used in this field

*We have adjusted this throughout the text to read as transferability or ease of applicability where more appropriate (also in response to Reviewer #2).*

- L113-L114: "As such, the transferability of near-surface air temperature models remain largely unknown. Analysis . . ."

*We have adjusted this in line with the reviewers comments.*

- L115-116: simplify: "In this study, we use new datasets of on-glacier air temperatures observed at three glaciers . . ."

*We have simplified this now using the reviewers suggestion.*

- L118: the word 'distributed' is probably unneeded

*Removed*

- Sec 3.3 can be moved up and included with the 3.1, which can be renamed "Meteorological observations", but just start with "Air temperatures (Ta), incoming shortwave,. . ."

*We have merged these sections now based upon the reviewer's advice.*

- L193-194: remove the first part: "Flowline distances (m) for each glacier were calculated with the TopoToolbox. . ."

*Done*

- L200: remove "between the aforementioned studies."

*Removed as suggested.*

- L206-207: Feels like this needs a more general introduction: "Our methods consist of (1) aggregating temperature observations based on off-glacier temperatures, (2) generating off-glacier temperature gradients to compare on and off-glacier temperatures, and (3) estimating the glacier cooling effect on near-surface temperatures by fitting parameters to the SM10 model." Your rationale can then be moved into the individual subsections.

*Thanks a lot for this. We followed the reviewer's advice and have included the suggested paragraph in the text.*

- L226: replace "within the glacier boundary layer" with "between ambient and on glacier temperatures"

*Changed based upon the reviewer's suggestion.*

- L226 – 229: this could be simplified to read "We calculate hourly ambient lapse rates from the off-glacier weather station and temperature loggers T1_94, T2_94, and T1_390. These sites are assumed to be unaffected by the glacier cooling effect."

*We have simplified this as suggested.*

- L234: ". . .T-logger to quantify the difference between ambient and on-glacier Ta at each site. . ."

*Adjusted*

- Eq 1: asterisks should be superscript in this equation, and see general comments above

*Adjusted*

- L276: suggest using 'transferability' instead of 'generalisability'

*Done*

- L284: is the ELR suitable for all sites and seasons?

*See above response to the reviewer's main comment.*

- L308: since there is a scale break in the figure, it is tough to see this as 'linear'

*We understand the reviewers point and now present the same figure (Figure 3) without the scale break. We intended to add clarity to the figure this way, but now remove it as information (about linearity etc) was potentially lost.*

- L349 – 351: Simplify! "For P90 conditions (Fig 6a), differences between TaAMB and observed on-glacier temperatures are up to 5.8 C at flowline distances greater than 7000 m."

*Adjusted*

- L351: avoid 'heighten' in this context

*We have reworded this to "increase" to be more clear that we refer to air temperature biases.*
- Sec. 5.2 : avoid using "offset"
*Adjusted to ' difference' or similar throughout*

- Figure 7: perhaps show these relations as scatterplots. Tough to interpret the relation using line and bar plots.

*We have now added subplots to show the relationship of Ta differences and SWIN as suggested by the reviewer.*

- L352 – 353: RH will typically be high when temperatures are low, so this is not surprising.

*We agree with the reviewer that this is not such a surprising result and we therefore do not explore it or discuss in detail . We adjusted the sentence:*

 *"This is generally associated with drier conditions, and for hours of greater relative humidity (AWS_Off), when conditions are generally cooler, differences are unsurprisingly smaller (Figure 6b)."*

- L382: "notably distinct" doesn't really fit with the results in the table, as the coefficients of SM10 are within the confidence intervals of the other datasets

*This is a good point, and we thank the reviewer for bringing it to our attention. The text has been adjusted accordingly to include this:*

 *"Accordingly, the exponential functions that are fitted to the observations at Parlung glaciers and those of the original study are distinct (red and blue lines in Figure 8, Table 3), although within the confidence intervals of each other."*

- L430: need to specify energy balance models. Temperature indexed approaches don't need this correction if the melt factors are calculated with off-glacier temperatures.

*Intermediate models, such as the Enhanced Temperature Index (ETI) however do calibrate to on-glacier observations… So while EB models are the key subject here, we include mention to ETI models too.*

- L474: "While the data from the Parlung catchment. . ."

*Done.*

- L493: southern Coast Mountains (not Canadian Rockies)

*Changed following the reviewer's suggestion.*

---

## Author Comment (AC2) · 4 Dec 2020

**Referee #2 - Luca Carturan**

*We would like to thank the reviewer very much for his valuable and in depth comments on our manuscript. Following his advice we have now made the following key changes to our manuscript draft (which we respond to specifically below):*

1) *We have adjusted the title to more appropriately reflect the content of our work, giving specific mention to the Tibetan Plateau case study and given the revised terminology (following comments of both reviewers).*
2) *Simplified the text following both reviewer's comments to be clearer and less ambiguous.*
3) *Added additional information and discussion regarding the challenges of not using artificially ventilated radiation shields and the expected impact on the study.*
4) *Added some additional discussion of the glacier width/length ratio in explaining the presence of increased temperature sensitivity on glacier termini, as suggested by the reviewer.*
5) *Adjusted the terminology and changed "climatic sensitivity" to "temperature sensitivity" throughout the manuscript (following the request of reviewer #1).*
6) *Changed and adjusted figures based upon the requests of both reviewers.*

General comments

The paper from Shaw et al. presents a new interesting dataset of on-glacier air temperature measurements, which are compared to local off-glacier measurements in order to estimate the climatic sensitivity along the glaciers, and its spatial and temporal variability. In addition, the authors compare their results with similar datasets collected around the world, making assessments on the generalizability of the Shea and Moore (2010) approach.
The new dataset is of relevance, also because it provides additional observations for long flowline distances, which have been undersampled so far. The results are interesting and substantially agree with former parameterizations, confirming their good generalizability. This is a nice intercomparison among different sites in different geographic areas, even if it is not the first as suggested the Introduction.

*Thank you very much for your general comments, and for providing such a concise and to the point summary of the novelty and strengths of the paper.*

In my opinion the paper is rather well written, even if there are parts that require rewriting to be clear. I refer the authors to the specific comments, but I also suggest a careful proof reading (sorry I am not a native speaker).

*We agree with the reviewer here and have tried to shorten and clarify the text throughout, as documented in detail below.*

One of my major concerns is related to the 'accuracy' assessment of the air temperature measurements. I have written a specific comment on this regard. In general, the authors compare measurement datasets collected in the last decades using rather different experimental setups, in particular for radiation shields

that range from aspirated research-grade ones, to simple passive-ventilation tubes opened at both ends. The authors are well aware of this issue, as they write in the discussion and the outlook. I am not criticizing the single experimental designs and approaches, but I agree with them that this is one of the main issue for intercomparisons, often overlooked, and that in the future this aspect requires higher attention.

*We would like to thank the reviewer for his valuable and in depth comment on this on our manuscript. As the reviewer states, we do highlight the key issue of sensor intercomparison in the discussion and completely agree with the reviewer that artificially aspirated radiations shields should be a priority in future research of this kind, even though it is unlikely to occur for a lot of the available data in such intercomparisons as this. We have followed the reviewer's advice and provide a greater discussion of this challenge moving forward.*

According to me, the authors should add some discussions on the influence of local glacier surface topography, and in particular the width/length ratio of glacier tongues (i.e. the heating from surrounding slopes that are free from ice cover) and the role of surface steepness upslope of Ta sites. Clearly, the Shea and Moore (2010) does not account for glacier slope, which however is important in regulating the local prevalence of the cooling caused by the loss of sensible heat over the adiabatic heating for descending air. It could be interesting to observe that the Greuell and Böhm (1998) approach accounts for this.

*This is an excellent comment, and we thank the reviewer for it. In response to it, we now: add a section in the discussion where we estimate the glacier width-length ratio as suggested by the reviewer and analyse its relationship to the observed increases in temperature sensitivity on glacier tongues. We have also added a figure to the supplementary material section that shows which glaciers show this observed temperature sensitivity increase (relative warming on the tongue) and what is their width-length ratio. We found that the width-length ratio indeed offers a potential mechanism for the observed increases in temperature sensitivity (see specific responses below) on some of the tested glaciers. Accordingly, we have now added more discussion regarding the role of longwave emission and glacier slope in the manuscript.*

Specific comments
Title: please consider this or similar alternatives (less generic): Distributed summer air temperatures across three glaciers in the Tibetan Plateau: climatic sensitivity and comparison with existing glacier datasets
*We agree with the reviewer and have provided a more specific title. In line with the comments of reviewer #1 regarding changing the terminology, we have changed 'climatic sensitivity' into 'temperature sensitivity' throughout the manuscript. Our revised title is thus:*

**Distributed summer air temperatures across mountain glaciers in the south-east Tibetan Plateau: temperature sensitivity and comparison with existing glacier datasets**

**A note on the above terminology**: *While climatic sensitivity has previously been used by studies of this type (including the reviewer's work), we follow the suggestion of reviewer #1 to avoid confusion with 'climatic sensitivity' being interpreted as an indicator of melt (e.g. mm w.e. °C) and re-term this 'temperature sensitivity'. We provide a definition of this in the introduction as being the ratio of on- glacier Ta to Off-glacier estimated Ta at the same elevation. We preferred this over the term 'cooling effect' suggested by reviewer #1 as this would not define accurately both the glacier cooling and dampening effect.*

A30: both the Ta 'on' (or over) the Tibetan glaciers
*We have changed this to 'on' following the reviewer's suggestion.*

31: In general, observations on small glaciers (with flowline shorter than 1000 m) reveal that they are highly sensitive to temperature changes outside the glacier boundary layer. Or maybe I have misunderstood the meaning, a possible alternative could be: In general, air temperature on small glaciers (with flowline shorter than 1000 m) is highly sensitive to temperature changes outside the glacier boundary layer.

*The reviewer did not misunderstand the meaning of our sentence. We follow the reviewer's suggestion and have adjusted this sentence accordingly.*

35: please rephrase, e.g. replace 'remain associated with other warm air' with 'is affected by warming' processess that increase…
*Changed now using 'affected by'.*

40: is still required to explain 'the variability' of these effects 'on' different glaciers
*Changed following the reviewer's suggestion.*

48-49: that or which?
*Sentence restructured following advice of both reviewers to be:*
*"A free-air ELR cannot be reliably used to estimate near-surface air temperatures above melting glaciers, where steep temperature gradients are found within 10 m of the surface under warm 'ambient' (off-glacier) conditions (van den Broeke, 1997; Greuell and Böhm, 1998; Oerlemans, 2001; Oerlemans and Grisogono, 2002; Ayala et al., 2015)."*

51: maybe better 'increases' rather than 'heightens'?
*Adjusted as above.*

74-76: please rephrase to improve readability of this sentence (and in general of the paper, adding some commas, or splitting long sentences)
*We have now split this sentence as "This approach considered the ratios of observed on-glacier temperature and estimated ambient temperature for the elevation of a given point on a glacier (hereafter 'T$_a$Amb'). The authors calculated two ratios from a piecewise regression above and below a critical threshold temperature for the onset of the glacier katabatic boundary layer (KBL - see section 4.3)."*

77-78: have been tested by …..
*Changed*

82-86: this period requires commas and/or splitting
*Now split and written as:*
*"This second, physically-oriented approach was developed by Ayala et al. (2015) to account for a relative 'warming effect' evident on the termini of some mountain glaciers when compared to upper elevations"*

88: adds, additional, is repetition ('additional' should be removed in my opinion, also because there is another one at the end of the period)
*Good point, 'additional' has been removed from the sentence.*

95: consider replacing 'Thus' with 'In this way'. The separation from the period above is questionable in my opinion
*Adjusted following the reviewer's suggestion.*

98: Because distributed on-glacier observations are often limited, and rarely cover the full length of the glacier boundary layer (or flowline?), this additional correction for warming associated to the unknown parameters (am I right?) of ModGB can lead to high uncertainty in Ta estimates at the glacier terminus (or maybe better 'lower part'?)
*The reviewer is correct that we refer to the unknown parameters of the ModGB approach. We have adjusted the sentence following the reviewer's suggestions as "Because the available distribution of on-glacier observations is often limited and rarely extends for the entire length of the glacier flowline, this additional correction for warming associated with the unknown parameters of ModGB can lead to high variability in T$_a$ estimates on the lower glacier ablation zone (Troxler et al., 2020) (Figure 1a). ".*

104: to be the cause of 'the' relative warming 'at' the glacier terminus

*Adjusted to:*

*"In contrast to this, the statistical method of Shea and Moore (2010) provides a simpler estimation that has fewer assumptions and parameters, though it does not explicitly account for physical processes that are thought to cause 'relative warming' on the glacier terminus"*

107-109: in my opinion, the simplicity and the statistical nature of an approach are not sufficient by themselves to make it more generalizable than more complex schemes. Maybe you meant 'easily applicable'? The generalizability of the Shea and Moore (2010) approach was instead assessed by previous works and applications to other glaciers. Please consider revising.

*We agree with the reviewer that the SM10 approach is more easily applicable due to its statistical nature. It is not necessarily better performing or more generalisable, and we do not show it in this study, despite the fact that some previous studies provided some evidence of this. Accordingly, we have reworded this to state that due to its relative simplicity, it is more easily applicable.*

119: on-glacier observations available around the world. (I would delete 'made to date').

*Removed*

111-112: I do not agree with this statement. Please see for example the comparisons made by Greuell and Böhm (1998) in their Figure 5, and by Carturan et al., (2015) in their figures 6 and 8

*This is a good point and an oversight on our part. We agree completely with the reviewer. We intended to state that it was the first to compare such a quantity of* distributed *air temperatures on glaciers at sites around the world (previously limited to Europe + Shea and Moore's study). However, we have reworded this now:*

*"To date, few studies studies have investigated the variability of distributed, on-glacier $T_a$ at different sites around the world…"*

130: 'wider Parlung catchment' is not so clear. Please see also my comments in this regard for Figure 2.

*Adjusted to Parlung-Zangbo catchment.*

137: and the 'longest' rain season? (I would remove 'annual')

*Agreed. Adjusted following the reviewer's suggestion.*

141: please clarify what is the 'more elevation-independent mass balance sensitivity' (lower sensitivity of mass balance to elevation?)

*Indeed. This was a term from the cited paper. We have adjusted following the reviewer's alternative suggestion. While glaciers in the drier interior of Tibet show a strong relationship of mass balance and elevation, this is not evident in the south-east Tibetan Plateau.*

142: 'Because Tibetan glaciers are shrinking and fragmenting, the accurate estimation of on-glacier temperatures is relevant for investigating and modelling their climatic sensitivity (Carturan et al., 2015). However, to date, no studies regarding the distribution of on-glacier temperature have been performed within the Tibetan Plateau'

*We thank the reviewer for their suggestion which we have added now.*

149: in this section you report the manufacturer and model of temperature sensors but information on manufacturer and model of radiation shields is of equal (or even larger) importance. Please add this information.

*This is a good point. We have added this information now. The manufacturers of the shields in our manuscript do not provide estimated uncertainty values, however, as other manufacturers (e.g. Campbell) do. Therefore we cannot provide an expected value of the uncertainty, though we know that this will be heightened under low wind, and high insolation conditions. Nevertheless, we show that the temperature values are comparable and do not deviate largely from the Vaisala T-RH probe and shield at the on-glacier AWS (metres away - section 3.2).*

156: from 12th July 'to' 18th September (exactly the same day/month in both years?)
*Yes, we filter the full period of T-logger measurements (12th July in 2018 until 18th September in 2019). We therefore compare two summers, limiting the comparison to that same period. This forms our 'common observation period' as written in the text.*

157: here you mention Table 1 after speaking about data gaps, but Table 1 does not report this information, please check or modify
*This is a good point and we thank the reviewer for identifying our mistake. Data gaps were very minimal and we subsequently removed that column from our table due to a minimal amount of information gain. We have now removed this mention from the text.*

163-164: the sentence from 'For' to 'station' leaves the reader with incomplete information on the spatialization of off-glacier air temperature. Is it in the following sections? If so, move this sentence there; if not, complete here
*We have now removed this sentence from this paragraph as it was already mentioned in section 4.2.*

169: in my opinion this intercomparison is useful, but it does not reveal the 'uncertainty' of measurements, which would require an intercomparison among instruments and shields under controlled conditions (see e.g.: https://www.wmo.int/pages/prog/www/IMOP/PastIntercomparisons.html). Instead, it enables to assess the 'comparability' of measurements, which is relevant in this work but is different from uncertainty (it would require at least a comparison towards an artificially-ventilated radiation shield). I want to highlight this difference because both T44 and AWS_On have passive-ventilation systems, which are subject to the same error under similar conditions. For example, they are likely to lead to significant overestimation of air temperature under high radiation and low wind speed. In these circumstances a small difference between the two instruments does not reveal that they are affected by small uncertainty, it only enables to state that they are comparable. I have measured differences up to several degrees in these conditions between passive- and active-ventilation systems on glaciers, especially with high surface albedo. This is the reason why I prefer using active-ventilation systems, even if I perfectly understand the advantage of using passive ventilation systems, which in harsh high-mountain environments are often the only option (also considering costs, power requirement and possibility of failure of active systems).

*We completely agree with the reviewers point about comparability vs uncertainty here and have reworded this throughout our text, and use this argument when discussing our findings. Indeed, it would be ideal to compare these measurements to the air temperature measurement standard that is artificially aspirated. As the reviewer clearly states, the usage of 'active' ventilation systems is often not possible due to power requirements and equipment failure and unfortunately was not possible for installation on the Parlung glaciers for this study. We note that while these measurements at two sites give a test of comparability and not necessarily uncertainty compared to a reference measurement, they also demonstrate how temperatures might deviate due to heating errors under high SWin, low-wind speed conditions (Figure S1). Nevertheless, for lower glacier ablation zones (i.e. at large flowline distances such as for this intercomparison), wind is almost always present (providing adequate ventilation to these sensors). At higher elevations, stations may be exposed to synoptic winds which also aid the ventilation of temperature measurements. However, if sheltered from wind, one might expect higher errors at those stations for short flowline distances. Unfortunately, we cannot test this with any of the given data available. We have given more consideration to these points in the text, following the reviewer's valuable suggestions.*

170: in table 1, T44 is at 4809 m (same elevation of T34), whereas AWS_On is at 4649 m. On the other hand, in Figure 2 T44 and AWS_On are at the same place, therefore I think there is an error in Table 1
*We thank the reviewer for identifying this error which has since been corrected to 4808 m a.s.l.*

171: that are located (remove 'co-')
*Removed following the reviewer's recommendation*

174: this 'P90' is not clear, it looks like an alternative name for the AWS_Off. Please clarify, for example you could write: '……temperature at AWS_Off. We find that for these warm hours (hereafter referred to as 'P90' - (Ayala et al., 2015; Shaw et al., 2017; Troxler et al., 2020)), when the KBL development is theoretically at its strongest (e.g. van den Broeke, 1997; Oerlemans and Grisogono, 2002), 95% of hourly differences……'

*We thank the reviewer for their suggestion which we have used now.*

183: based on these results, did you consider filtering the temperature data, e.g. excluding those with ERA5 wind speed lower that 2 m/s?

*We did also filter results by low wind speeds for Parlung4 (with available wind speed data on the lower ablation zone), but found no major differences in the results of 'temperature sensitivity' (i.e. k1 and k2). We note that we compare ERA5 wind speeds for all glaciers (Figure 6) due to the absence of wind speed measurements at Parlung94 and 390. However, filtering all results by ERA5 wind speeds would poorly characterise the temperature differences/temperature sensitivity in this study. ERA5 does not seem the appropriate dataset for this. This is because the highest synoptic (ERA5) wind speeds correspond to generally cooler conditions, whereas on-glacier, the higher wind speeds correspond to the hours that are warmest and with clear sky conditions due to katabatic winds (evident from the AWS_on records for Parlung4). Accordingly, the conditions that promote the highest sensor errors rarely occur at the same time (warmest, clear sky conditions == higher wind speeds - see Figure 5). For the few occasions that did correspond to both warm, clear sky and calm hours, temperature differences were > 1°C. While these differences may have been larger if tested against a reference, artificially ventilated measurement as the reviewer mentioned, filtering them from our analysis did not change our main findings.*

186-188: 'In addition to Ta, in this work we used incoming shortwave radiation and relative humidity measured at AWS_Off, on-glacier wind speed measured at AWS_On, and 'free-air' wind speed and direction derived from ERA5 reanalysis data (C3S, 2017)'.

*Following the suggestions of reviewer #1, we have now restructured this section and incorporated it into 3.1.*

192: to obtain elevation information

*Reworded now to say 'obtain'*

201: please remove 'that' before 'the impact'

*Now adjusted*

202: at few 'selected' points

*Changed following the reviewer's suggestion.*

206: 'for aggregation of on-glacier ta data into sub-groups, and for calculating the distribution of off-glacier ta in space.

*We have now reworded the start of this section following reviewer#1's comments to say:*
*Our methods consist of (1) aggregating temperature observations based on off-glacier temperatures, (2) generating off-glacier temperature lapse rates to compare on and off-glacier temperatures, and (3) estimating the glacier cooling effect on near-surface temperatures by fitting parameters to the SM10 model."*

209: and 'a quantification of' the effect of the glacier bl. (is referred to the 'allows' above)

*Now restructured and simplified as above.*

212: are 'described' in sections…

*Changed*

214: please state explicitly to what P10 and P90 correspond, i.e. the 10% coldest and the 10% warmest off-glacier temperature? and what about data between them?
*Now added to this sentence*

217-219: unclear
*Now written as: "...we bin all contemporaneous observations of on-glacier $T_a$ at each T-logger that correspond to the same hours as the coldest (P10) and warmest (P90) observations at AWS_Off"*

219-222: can be written more clearly, e.g. add 'on-glacier' to Ta (be explicit)

*Now written as: "We evaluate how strong the linear relationship of on-glacier $T_a$ with elevation and flowline distance is for these subgroups using the coefficient of determination ($R^2$)."*

234: consider removing 'bias'
*In line with comments and changes based upon reviewer #1, we have adjusted the text in this section to read:*
*"We consider this as the best available approach to estimate the ambient lapse rate for the catchment. We compare the hourly estimates of extrapolated off-glacier $T_a$ ($T_a$Amb) with the observations at each on-glacier T-logger in order to i) quantify the differences and how they relate to meteorological conditions and glacier flowline distance; and ii) parameterise the along flowline temperature sensitivity to $T_a$Amb following Shea and Moore (2010) (section 4.3)"*

278-279: I do not understand this sentence from 'we' to 'site' (maybe a repetition of what is reported at L286-287?)
*We apologise that this sentence was ambiguous. We referred to hours at each glacier study site (Parlung, Arolla etc) where all available stations were available. "We sub-group data for each glacier to those hours during the summer when all on-glacier observations were available."*

299: what is 'the best available value from the relevant literature' for the annual precipitation at the AVDM site? Carturan et al., (2015) report 1233 mean annual precipitation at that site from 1979 to 2009, therefore I wonder where the authors found 784 mm as reported in Table 2 (it looks like uncorrected raw data from the Careser diga weather station….). In 2010 and 2011 the annual precipitation was above average (i.e. about 1400 mm).
*Our apologies to the reviewer that we had misreported their precipitation value here. We have now added the 1979-2009 average as reported by the reviewer in their paper, though with mention in Table 2 to the above average years of 2010-2011.*

308: please report n for all hours
*Now added (n = 3312).*

309: flowline 'distance' (or DF), also in the following
*Updated now*

312: and 'with' flowline distance
 *Added now*

321: when comparing 'Ta' to glacier elevation or flowline (distance)?
*Added following the reviewer's suggestion.*

324: please add 'at the same elevation' after 'Ta'
*Now added*

335: 'i.e. where on-glacier observations are expected to match Taamb'

*Updated following the reviewer's suggestion.*

340-342: I understand the meaning but this sentence should be written more clearly, e.g. there is a good alignment between on/off-glacier temperature until the katabatic winds start to blow on Parlung4

*We agree with the reviewer and have simplified this now as "On-glacier $T_a$ and $T_a$Amb align well until the onset of katabatic winds (on Parlung4 and only assumed for the other glaciers due to lack of on-glacier wind observations – Figure 5)."*

345: and 'extension' of the katabatic wind into the proglacial area

*Adjusted*

347: also here consider removing 'bias' (in any case I would use bias or offset, but not both). In the following you use only offset so I would stick to that

*We have adjusted this to be the difference of on- and off-glacier Ta, following the other reviewer suggestion not to use offset.*

351: These effects appear to 'increase' beyond 2000 m 352 along the flowline on the Parlung94, though 'significant' offsets are observed for all glaciers (and sites?)

*We had referred to all glaciers, but not necessarily all sites. We have now adjusted to read: "These differences appear to increase beyond 2000 m along the flowline (Parlung94), though significant differences can be witnessed for all glaciers (different symbols in Figure 6)."*

360: would it be possible to state that, besides individual meteorological variables, the offset is largest for warm/anticyclonic conditions, and lowest for cool/cyclonic conditions?

*We thank the reviewer for this suggestion. It is an appropriate description of the findings and we have added it to this section.*

364: the terminus 'of' each glacier

*We have adjusted this mistake now.*

366: recorded at AWS_off

*We thank the reviewer for the correction.*

369: though 'it' varies….

*Changed now.*

371: '…September, while the offset for Parlung4 and Parlung94 remain significant'. Possible explanation for that?

*Good point. It is possible that over smaller flowline distances (as for Parlung 390), only the very warmest conditions promote a sufficient enough cooling and dampening effect. For larger flowline distances, there is a katabatic wind layer generated over sufficient fetch that it continues to produce strong 'cooling' relative to the ambient off-glacier Ta, even under less warm (not P90) conditions. We have elaborated slightly on this in the text, "For 2019, maximum daily $T_a$ offsets on Parlung390 steadily increase during July and August then fall close to zero in September. The bias offsets for Parlung4 and Parlung94, however, remain sizeable (Figure 7), perhaps due to the persistence of katabatic winds over a larger flowline distance even under relatively cooler conditions of September."*

376-387: this part is written rather poorly, please improve it (concepts are more clear and better described e.g. in the caption of Figure 8). In particular, try to avoid writing 'parameterisations' (too general) and write explicitly whether you are referring to 'sensitivities' or 'fitted exponential functions'

*We have now attempted to shorten this paragraph and reword it following the reviewer's advice. We have now referred to exponential functions as suggested by the reviewer:*

*" Accordingly, the exponential functions that are fitted to the observations at Parlung glaciers and the original study are distinct (red and blue lines in Figure 8, Table 3), although within the confidence intervals of each other. Fitting an exponential function for all sites where a down-glacier decrease in temperature sensitivity (k2) is evident (black dashed line in Figure 8b) clearly misrepresents many of the observations, particularly those at greater flowline distances, balancing the behaviours reported for different sites. "*

388-396: these results are interesting, and I wonder if there can be some influence from the very low width/length ratio of the tongues of McCall and Juncal Norte, surrounded by ice-free slopes that are expected to warm considerably… Parlung4 instead has a more 'compact' shape (i.e. less thin and long tongue). In addition I ask the authors to consider possible differences in the sensors and radiation shields used in the various referenced works and datasets (e.g. the radiation shields used in Juncal Norte were PVC cylinders opened at both ends, that are very different from the generally-used multi-plate radiation shield)

*This is a very interesting idea and we thank the reviewer for suggesting it! While we believe that the very distinct behaviour for Juncal Norte (Fig 8) is a result of the up-glacier wind during the warmest hours (Pellicciotti et al., 2008; Petersen and Pellicciotti, 2011), the reviewers' suggestion of valley confinement and the influence of warming slope could have an influence. We measured an approximate width of each glacier from Google Earth (not on the exact date for each dataset, but still providing a general idea), taking the average width over the lower ~30% of the glacier (i.e. Figure 10). We show below (in Figure R1) the ratio of this width to the total glacier length, where the coloured boxes are those sites grouped as showing increased temperature sensitivity (as in Figure 10). It suggests that many sites with thin tongues have evidence of this increased sensitivity (as suggested by the reviewer), though there is not a clear pattern for those sites without the increasing temperature sensitivity (hollow boxes in the figure below). The degree to which the glacier tongue is constrained by the valley may likely play a role through longwave emission as well as controlling exposure to synoptic winds etc. We have now included more discussion on this in the manuscript as well as adding the figure below to the supplementary material.*

*We now adjust our discussion in the text regarding the different sensors and radiation shields considered by those studies and how they might impact the findings of the current study.*
*"Equally, the uncertainty of the actual observations (e.g. section 3.2) is hard to clearly define due the variable instrumentation (sensors and radiation shielding), on-glacier location and local topographic and micro-meteorological effects of each study site (Table 2). Further work on a unified model of estimating $T_a$ should need to address these issues, perhaps with further, dedicated analyses. Because our study, and many similar studies of this kind, did not have artificially ventilated radiation shields available, the uncertainty of the measured $T_a$ is difficult to quantify. We consider this to be less problematic at large flowline distances, where good ventilation to the sensors is often provided by the glacier katabatic wind layer even under warm conditions. However, at short flowline distances in the glacier accumulation zones, uncertainty of both the on-glacier observations and ambient $T_a$ extrapolation is larger. Artificially ventilated radiation shields are not commonplace in glaciological research due to the additional power demands that often cannot be met, though would be strongly encouraged for further research into the temperature sensitivity of mountain glaciers."*

[Figure]

*Figure R1: The calculated width/length ratio of each glacier (Bridge = 'CMBC', La Mare = 'AVDM') compared to the presence of increasing temperature sensitivity on the glacier terminus (coloured boxes). The glacier width was calculated from Google Earth imagery as an average of the lower 30% of the glacier.*

399: the investigated sites lie close to the original SM10 exponential function up to ~4000 m….
*Adjusted following the reviewer's suggestion.*

416-423: poorly written, please clarify (requires multiple reading)

*We have re-written and improved this paragraph.*

427: bring additional evidence of the spatial…
*Adjusted based upon the reviewer's suggestion*

429: and highlights the need to appropriately account for these effects in glacier melt models
*We thank the reviewer for identifying an error in our writing… this had been deleted somehow from the text. Now it is added again.*

431: remove 'since'?
*Removed*

440: To the authors' knowledge?
*We thank the reviewer for identifying this error.*

487: remove 'more' before 'generally following…'
*Removed now.*

489: 'in agreement with the 'ModGB' model…'
*Adjusted following the reviewer's suggestion*

499: the most distant station: the station with the largest flowline distance?
*Yes. We have now changed this to specify the station at the largest flowline distance.*

564: for a more limited number of total observations: do you mean 'from a smaller sample size of temperature observations'?
*Yes. We have reworded this following the reviewer's alternative suggestion.*

566: however we found negligible sensitivity of derived K2 on T* (If I understand the meaning correctly)
*We do indeed refer to the negligible sensitivity of derived k2 and have reworded this accordingly.*

575: rise or onset or increasing strength? I would rewrite, e.g.: However, ignoring the differences in climatic sensitivity that have been reported before and after the onset of the KBL, is arguably an over-simplification and does not enable to describe correctly the observed behaviour
*We have re-written this following the reviewer's advice.*

578-584: poorly written
*We have followed the reviewer's suggestions and attempted to improve the writing in this paragraph.*

588: please remove 'more' after exploring. Carturan et al. (2015) used these ventilated devices, are there other studies with ventilated shields among those compared in this work? I would add here the related references

*Adjusted as the reviewer suggested, and we have now cited the reviewer's paper. Indeed, few other studies had artificially ventilated radiation shields for temperature measurements, or they haven't been explicitly reported if so.*

591: 'at' the glacier terminus
*Removed 'the'. We refer to several glaciers, thus in plural.*

593: for certain glaciers or glacier areas?
*Changed to areas.*

606: please replace 'are more climate sensive' with 'display high climatic sensitivity'
*Changed to display high temperature sensitivity (see note on terminology change).*

607: that is close to 1 (remove 'r' at the end of closer, otherwise you should report the term of comparison)
*Changed now.*

618: compared to the existing parameterization
*Changed now following the reviewer's suggestion.*

623: is associated with (not 'remain')
*Changed to 'is'*

624: I would add (also in the discussion) the thermal (infrared) emission from the surrounding bare rocks, especially for thin and long ablation tongues
*We have now added a paragraph to give consideration to this idea (see above) and mentioned it here.*

627: this is not true if the tongue is thin (small % area compared to the total glacier area). I would rewrite: is highly important because this is the glacier part th.at is most affected by ablation
*Now re-written as: "The terminus of some glaciers is associated with other warm air processes, possibly due to boundary layer divergence, warm up-valley winds or debris cover/valley heating. We find that these*

*effects are evident only beyond ~70% of the total glacier flowline distance, although further work is required to explain this behaviour. A better understanding of temperature variability for this lower 30% is highly important as this part of the glacier is most affected by ablation."*

633: to enable the construction of a generalised model for on-glacier air temperature estimation.
*We thank the reviewer for their alternative recommendation which we have added now.*

Figure 1: in my opinion the green parts of fig. 1a can be removed. I would also change the colour of (a), (b) and (c) from red to black, and try to uniform their placement (e.g. upper right corner of their frames). I would also shorten (and make more easily readable) the caption. The difference between (b) and (c) is not well explained. In a general scheme like this, I think that one should also find a short explanation of what k1, k2 and the yellow star represent
*We have followed the reviewer's suggestions, including shortening the text and explain better the differences between b and c).*

Figure 2: in b) I cannot see the map of a catchment, i.e. one expects to see the boundary of a hydrological catchment, with its divide, rivers and outlet. Please rephrase. When you mention suffix please refer to table 1. Please note that (A) does not show elevation data (it is (b)). c-e show the calculated flowline distances. Please consider homogenizing the scales for flowline distances. Off-glacier T2 on Parlung4 and T3 on Parlung94 actually look like 'on-glacier' in the figure.
*We thank the reviewer for identifying the mistakes in our caption that was not updated after adding the regional map (actual panel 'a'). We now change our definition of catchment for Figure 2b. Regarding the scales, homogenizing makes it more difficult to interpret the short flowline distances (e.g. for Parlung 390) which are relevant to the manuscript, and so have kept the figure as it was. The GPS points for the $T2_4$ and $T3_{94}$ are very close to the glacier terminus but actually off-glacier, as stated in Table 1.*

Figure 3: caption 'The mean Ta against elevation…' (see caption Figure 4)
*Adjusted following the reviewer's suggestion.*

Figure 4: title of second chart (P90) is partially cut. Caption: 'The red line indicates the piecewise lapse rate above the elevation of T1_390 to lapse Ta to the top of the flowline'
*We thank the reviewer for identifying this issue which was the result of overlapping legend and titles in the Matlab Figure export. This has now been adjusted.*

Figure 5: please write explicitly 'wind speed' in the y axis (WS is the acronym but generates confusion with the WS of 'AWS' to a quick reader). Caption: you can remove the sentence from 'No' to 'data', it is sufficient to describe it in the maintext.
*We have updated the figure and caption based upon the reviewer's request.*

Figure 6: why are the legends semi-transparent? Caption: (estimated 'from Taamb' - observed). high 'is' > 2.5 m s-1 and low 'is' < 0.7 m s-1)
*The semi-transparency is an artefact of the scale break plotting in Matlab that we could not circumvent. This has now been recoloured in an external process for the final figure. Captions have been adjusted.*

Figure 7: in the caption: Maximum daily Ta offsets (estimated 'from Taamb' - observed) at the on-glacier T-Logger closest to the terminus of each glacier for 2018 (missing!) (top) and 2019 (bottom).
*We thank the reviewer for identifying our mistake which has now been corrected.*

Figure 9: Caption: The original SM10 parameterisation is retained in the top panels
*Added.*

Figure 10: The k2 sensitivity along the normalized flowline compared to the total glacier length (colour bar). Glaciers have been grouped in two clusters: a) those with downglacier decreasing sensitivity, and b) those with increasing sensitivity towards the glacier terminus.
*Suggestion now added to the caption.*

Figures 9 and 10: I find difficult to refer to Figure 8 for site coding (consider repeating the legend with geometric symbols in these figures)
*We have now added the legend to Figure 10 to make it simpler for the reader, rather than referring back to Figure 8. We could not find a way to neatly present this for Figure 9, however, so choose to retain our original Figure without this addition. We hope that it is deemed acceptable by the reviewer.*

Table 2: please check the precipitation data (e.g. AVDM and the 12000 mm in Chile). At line 981: 'Annual' precipitation totals….. Moreover, to which elevation are referred the MSAT and PT? I think it is relevant considering the high vertical lapse rates for both variables.
*We have adjusted the errors in the table and made a note of the precipitation for AVDM (see earlier comment) and a note to say that we adjust the MSAT to the mean glacier elevation as a general representation of the climatology for the site. We agree with the reviewer that high vertical lapse rates exist, though exploring them with ERA5 is beyond the purpose of this study, especially given the fact that they would be little representative of local conditions around glaciers.*

984: and those 'fitted'
*Adjusted now*